# Disadvantaged Economic Conditions and Stricter Border Rules Shape Afghan Refugees’ Ethnobotany: Insights from Kohat District, NW Pakistan

**DOI:** 10.3390/plants12030574

**Published:** 2023-01-28

**Authors:** Adnan Ali Shah, Lal Badshah, Noor Khalid, Muhammad Ali Shah, Ajmal Khan Manduzai, Abdullah Faiz, Matteo De Chiara, Giulia Mattalia, Renata Sõukand, Andrea Pieroni

**Affiliations:** 1Department of Botany, University of Peshawar, Peshawar 25120, Pakistan; 2Department of Botany, Islamia College Peshawar, Peshawar 25120, Pakistan; 3Department of Environmental Science, COMSATS University Islamabad, Abbottabad 22060, Pakistan; 4University of Gastronomic Sciences, 12042 Pollenzo, Italy; 5National Institute for Oriental Languages and Civilizations, 75007 Paris, France; 6Department of Environmental Sciences, Informatics and Statistics, Ca’ Foscari University of Venice, 30170 Venice, Italy; 7Department of Medical Analysis, Tishk International University, Erbil 4001, Kurdistan, Iraq

**Keywords:** ethnobotany, refugees, wild food plants, medicinal plants, ethnoveterinary, local ecological knowledge

## Abstract

The study of migrants’ ethnobotany can help to address the diverse socio-ecological factors affecting temporal and spatial changes in local ecological knowledge (LEK). Through semi-structured and in-depth conversations with ninety interviewees among local Pathans and Afghan refugees in Kohat District, NW Pakistan, one hundred and forty-five wild plant and mushroom folk taxa were recorded. The plants quoted by Afghan refugees living inside and outside the camps tend to converge, while the Afghan data showed significant differences with those collected by local Pakistani Pathans. Interviewees mentioned two main driving factors potentially eroding folk plant knowledge: (a) recent stricter border policies have made it more difficult for refugees to visit their home regions in Afghanistan and therefore to also procure plants in their native country; (b) their disadvantaged economic conditions have forced them to engage more and more in urban activities in the host country, leaving little time for farming and foraging practices. Stakeholders should foster the exposure that refugee communities have to their plant resources, try to increase their socio-economic status, and facilitate both their settling outside the camps and their transnational movement for enhancing their use of wild plants, ultimately leading to improvements in their food security and health status.

## 1. Introduction

One of the most promising trajectories in current and future debates in ethnobotany concerns the temporal and spatial changes in local ecological knowledge (LEK) and the diverse socio-ecological factors that may affect these dynamics. For this reason, the study of migrants’ ethnobotany can help to reveal some general human ecological concepts underpinning the universal phenomenon of human relocation. Worldwide, 89.3 million people are forcibly displaced, of which 27.1 million are refugees [1]. Whenever refugees leave their homeland and reach a new location they are challenged by the establishment of new relationships with the novel natural and socio-cultural environments, as well as with access to available food and health resources in the host country [2]. Often, migrant communities neither completely reject the majority culture’s ethnobotanical knowledge nor totally give up their own traditional practices, but rather find a compromise in the continuum between strengthening their own identity and fully adapting to the host culture [3]. The majority of refugees, especially those vulnerable in both legal and socio-economic terms, tend, however, to continue adopting their own medical help-seeking strategies, especially those concerning medicinal plants [4].

During the last few decades, a number of studies have explored the effect of forced or non-forced relocations on wild plant use and associated LEK. Indeed, changes in plant availability and accessibility among the migrated communities alter their own corpora of LEK [3,5]. For instance, different ethnic, linguistic, and religious groups in Azerbaijan retain different uses of wild food plants compared to the dominant Azeri population, possibly due to their different cultural background and their social–economic marginalization [6]. In addition, customs related to medicinal plants used by certain relocated populations undergo complex changes after migration; for instance, Colombian refugees and asylum seekers in London went through complex adaptation processes toward the new healthcare practices [4]. On this matter, some authors (among them [7]) have argued that sometimes the health policies of host countries substantially follow assimilationist principles, according to which migrating communities should adopt the ideas and habits of the majority group of the host society. Recently, some authors of this paper [8] suggested that the prolonged stay of Afghan Pathans in refugee camps in Mansehra District, Pakistan may lead to the erosion of their LEK related to wild food plants and herbs, as exposure to natural resources is very restricted in the camps. In order to further investigate this issue, we decided to look in more detail at the overall ethnobotany of Afghan refugees in Kohat District, NW Pakistan. 

Pakistan is one of the top five countries for number of hosted refugees (1,491,070 in 2021) [9] and thus it represents an ideal arena to address the changes in LEK related to wild plants held by refugees. Moreover, in NW Pakistan diverse peoples co-exist; for instance, local Pathans (Pashtuns), who have inhabited the area for at least a few centuries, live together with those Afghan Pashtuns forcibly relocated from Afghanistan to Pakistan between 1979–1981 because of the internal turmoil of the former country and the subsequent Russian invasion. During those years, different camps were built in Pakistan for Afghan refugees; moreover, in the late 1990s, some Afghan refugees started to move out of camps and settle in urban and peri-urban areas. 

The main objective of the study was to gain a better understanding of the trajectories of folk knowledge related to the entire spectrum of wild plants for food, as well as human and veterinary medicine, among Afghan refugees who arrived in Pakistan forty years ago and local Pathans. The specific objectives of the study were:To document the use of wild food and medicinal (human and veterinary) plants by local Pathans and Afghan refugees living inside and outside the camps in the Kohat District of NW Pakistan;To identify commonalities and differences in the folk plant uses among the three groups and to possibly interpret these findings in cultural terms.

## 2. Results

### 2.1. The Ethnobotany of Pathans and Afghan Refugees

We recorded the use of 145 wild plant taxa and four mushrooms belonging to 63 botanical families. Sixty-four percent of the recorded species were used as medicines, fifty-six percent as food, and twenty-one percent as veterinary applications (Table 1). In addition, a few plants were used as fuel wood (7%) and as cosmetics/fragrances (3%).

The highest number of taxa was reported by Afghan refugees living outside the refugee camps (AO; 73%), followed by Afghan refugees living inside the refugee camps (AI; 66%), while the lowest number of taxa was reported by local Pathans (48%) (Table 2).

### 2.2. Wild Food Plants

Our results show that most of the recorded wild food plants (WFPs) are used as raw snacks (39 taxa; 26%) or cooked as vegetables (38 taxa; 25%), while a few of them were reported as used both as raw snacks and cooked as vegetables, e.g., *Allium carolinianum* and *Solanum nigrum*. About 3% of the food uses were reported to be practiced in the past, while the remaining uses are still practiced by the studied communities. The highest number of plant taxa was mentioned by AO (*n* = 61), followed by AI (*n* = 46). *Apteranthes tuberculata* was the most common plant taxon among all the groups, although *Allium rosenbachianum* was very important among AI and *Mentha longifolia* among PO. Cooked preparations and raw snacks were equally important among refugees (30 and 29 among AO, and 25 and 26 among AI, respectively). Conversely, among PO, wild plant-based cooked dishes were far more common than raw snacks (28 vs. 14 taxa, respectively). All the other preparations were secondary.

Most of the WFPs are cooked in a very simple way: washed, cut, boiled, and fried in oil with onion, tomatoes, and chilies. Some of the reported WFPs are boiled and cooked with a combination of other wild vegetables according to taste preferences, i.e., *Atriplex laciniata* and *Chenopodium album* are boiled, fried, and cooked together among PO. Among the reported taxa, five taxa were utilized raw as salad items, i.e., *Apteranthes tuberculata, Nasturtium officinale*, *Thymus*, and *Malva neglecta* (mentioned by all the groups), and *Atriplex laciniata* (mentioned only among PO). A few taxa, e.g., *Achyranthes aspera* (used only by PO), *Thymus* sp. (mentioned by AO), *Myrtus communis* (mentioned by refugees), and *Olea europaea* (mentioned by all the groups), are utilized in making tea which has some therapeutic value. Only three plants were reported to be used to make sauces: *Allium carolinianum* and *Oxalis corniculata* among refugees, and *Mentha longifolia*. Among the wild vegetables, the fruits of *Carum carvi* (all groups), *Cuminum cyminum* (AI), the aerial parts of *Ferula foetida* (AO), *Mentha longifolia* (all groups), and *Trigonella foenum-graecum* (AO and PO) among the wild vegetables are predominantly exploited for seasoning.

Knowledge related to wild food plant preparation is vertically transmitted mainly from mothers to daughters.

### 2.3. Wild Medicinal Plants

Ninety-six taxa were reported to be used against ninety ailments. In general, digestive disorders were reported to be treated by the highest diversity of plants (32 taxa), followed by diabetes with 24 plants, respiratory disorders with 20 plants, fever and cough with 16 plants, and skin disorders with 12 plant species, while a maximum of 40 diseases/ailments are documented as treated by a single plant taxon. Nearly 7% of the medicinal uses were reported to be practiced in the past, while the remaining uses are still practiced by the studied communities. The results show that the majority of wild medicinal plants (WMPs) were used to treat the most common or day-to-day problems, i.e., fever, cough, stomach-related issues, itching, flu, chest pain, and internal and external wounds. It was reported that the communities also use the plant’s ingredients for the treatment of some major and severe ailments such as arthritis, tuberculosis, heart-related problems, ulcers, malaria, typhoid, hepatitis B and C, paralysis, and sexual disorders.

The AI group used the highest number of WMPs (*n* = 67) followed by AO (*n* = 58). *Artemisia scoparia* and *Berberis lycium* were the most common WMPs among refugees (AI and AO). *Withania coagulans* was reported by all the groups, but especially by local Pathans. Fever and diabetes were common ailments treated with WMPs among all the groups. Stomach problems and cough were also commonly treated with wild plants among AI.

*Zygophyllum indicum* is used to treat the highest diversity of problems (12 ailments), followed by *Berberis lycium* and *Nepeta laevigata* which are used to treat 11 ailments each. According to the study participants, *Berberis lycium* is thought to cure almost all types of ailments, such as digestive, respiratory, excretory, skeletal, and dermatological problems, as well as internal and external wounds. The majority of plants (41 species) were documented to have more than one part that is therapeutically important.

The majority of the medicinal formulations were administered both internally as an oral medication and externally as topical doses (skin, dental, and eye problems). Most of the remedies were prepared from a single species, while in a few cases, the preparations were a mixture of different plant species. For example, the root extract of *Melia azedarach* and the bark extract of *Juglans regia* were mixed and applied to the hair to permanently dye it black among refugees. The leaves of *Nerium oleander* and *Fegonia indica* are crushed and the extract is used against pimples and itching by AO and PO, while the leaves of *Peganum harmala*, *Trigonella foenum-graecum*, and *Lepidium sativum* are mixed and boiled, and the water is drunk to start menses in women among AI. In addition, AI boiled the seeds of *Artemisia absinthium* in water to soften them, after which they were given to children to cure the flu. Refugees mentioned that in the past adults chewed the seeds to treat the same problem. Likewise, the fruits of *Juniperus communis* were boiled in water, but only the water is given to children to treat fever, while elderly individuals consume both the fruits and water for the same problem. Overall, the common modes of preparation consisted of juice, paste, decoction, powder, infusion, and chewing raw plant parts.

Vertical transmission from mother to daughter was the most common way of sharing knowledge of wild medicinal plants. Nevertheless, it is men who harvest wild plants regardless of their final use.

### 2.4. Wild Veterinary Plants

We recorded 31 species concerned with the veterinary domain (plants used both for treating and feeding animals, see Table 1). More than half of the reported ethnoveterinary taxa (20 taxa) were used for feeding purposes. A maximum of 26 diseases or ailments were documented to be treated by the reported taxa, with the most common diseases being diarrhea, internal and external body wounds, constipation, intestinal worms, skeletal disorders, bloating, foot and mouth disease, fever, energy production, galactagogue, and urine blockage.

The top used wild veterinary plants (WVPs) were different among the three groups (*Berberis lycium* among AO, *Nepeta laevigata* among AI, *Tamarix aphylla* among PO), yet most of the WVPs were used for fodder and, to a lesser extent, wounds, intestinal worms, and diarrhea.

According to the study participants, most often ethnoveterinary knowledge is learned from elderly individuals and is passed to the next generation from father to child, but, in certain cases, it may be learned from friends or neighbors through general group discussions. The WVPs are often used in a very simple way, mostly through ingestion or application directly to the affected area. Decoctions are directly taken, or powders are mixed with milk or jaggery (a traditional non-centrifugal cane sugar) and given to the animal. Inside the plant body of *Calotropis procera* (mentioned by AI), a certain animal species (i.e., *Poekilocerus pictus*). These are dried, powdered, and put in the nose of animals (mostly cows and buffalos) to kill brain worms (any parasitic, worm-like species that inhabits the brain of another organism). In the past, the leaves of *Juniperus communis* were placed on embers by refugees and the smoke was inhaled by animals, shut in a closed room with the smoke for nearly half an hour, to remove intestinal worms. The decoction of *Berberis lycium* was reported by the majority of the informants in all the groups for healing internal and external wounds. One AI participant commented about the importance of *Berberis lycium*:

“Once the foot of my goat was broken. I gave her the decoction of Ziar largai (Berberis lycium) for 15 days and she recovered. A year later when she reached the age of sacrifice [the practice of the slaughter of animals that Muslim do in the way of Allah to earn reward, when animals reach nearly one year of age] and we removed its skin, we found a hard shell of Ziar largai around the broken bone”.

Nearly one-fourth of the ethnoveterinary uses were reported to be practiced in the past, while the remaining uses that are still practiced by the studied communities also have very low frequencies of quotation, which shows that the folk ethnoveterinary knowledge has eroded dramatically. The majority of the past ethnoveterinary uses were reported by AI, possibly due to the fact that inside the camps most families do not keep domestic animals on a regular basis. Only a few families were found to have animals (not more than two) in their homes due to poor economic conditions and limited availability of places to live. Both AI and AO mentioned that most of the time they were not allowed to freely herd their animals in the mountain regions, and therefore for this purpose they need to pay money to PO communities to rent a piece of land on an agreement for their herds, but most often it becomes difficult economically for both of the communities. The results show a dramatic loss of knowledge regarding ethnoveterinary practices due to the socio-economic shift occurring in the local communities as the majority of the inhabitants have decreased animal breeding and have abandoned pastoral activities.

### 2.5. Other Considerations on the Use of Wild Plants (Habitat, Gender, Plant Material Preservation)

More than half of the reported taxa mentioned by local Pathans are foraged in the fields near their houses as they are closely associated with agricultural practices and live mostly in plain areas, while the other two studied groups reported those taxa which are mostly available and foraged in mountains due to their geographical location and climate in their homeland in Afghanistan. The results show that 38% of wild taxa are foraged in Afghanistan, while only 19% of wild taxa are foraged in Pakistan; the remaining 42% of wild taxa can be foraged in both countries.

The wild plant taxa were collected from different habitats, e.g., fields, mountains, foothills, water banks, home gardens, graveyards, and sandy areas (Figure 1).

The majority of the reported taxa (36 taxa) were collected both from fields and mountains, while the remaining taxa were collected from water banks (14 taxa), foothills (11 taxa), graveyards (4 taxa), gardens (3 taxa), and sandy areas (2 taxa: *Citrullus colocynthis* and *Tamarix aphylla*).

Mountain environments were the most common foraging habitat for refugees (62 taxa among AI and 69 among AO), followed by plain fields (50 taxa among AI and 61 taxa among AO). The proportion of common foraging habitats is very different among local Pathans, as fields were the most frequent (58 taxa), followed by mountains (25 taxa).

One-fifth of the wild taxa were reported by more than 40% of the informants. The majority of the informants stated that the growth and availability of wild plant taxa were heavily affected by environmental and anthropogenic factors. The availability of certain plants, i.e., *Cichorium intybus*, *Tulipa* sp., *Apteranthes tuberculata*, and *Asparagus officinalis*, were reported to have decreased as a result of the low availability of water and precipitation rate in the study area. Conversely, the availability of certain plants, i.e., *Carum carvi*, *Punica granatum*, *Ephedra intermedia*, and *Nepeta laevigata*, were reported to have increased as a result of government restrictions to high elevation areas for security reasons, as these areas are considered unsafe and vulnerable to acts of terrorism. Because of overexploitation (for both human and animal consumption), the availability of *Zygophyllum indicum*, *Withania coagulans*, and *Prosopis juliflora* was reported to have decreased.

Foraging is mostly performed by male family members in Pakistan, while in Afghanistan both men and women are involved in foraging, possibly due to the fact that the camp environment is not suitable for foraging as different families belonging to different tribes live alongside each other which prevents women from engaging in this activity as it is against Pathans culture for women to come out of their houses and freely move about in front of male community members. In rare cases, in Pakistan, women who live outside the camps also forage for those species which can be found near their houses. The younger generation does not show any interest in foraging as they are mostly involved in small-scale businesses or work as laborers because of their low economic position and give most of their attention to modern and market-based products. One of the AO study participants commented:

“We don’t find enough time for foraging after returning from our work, and also give attention to those products which are easily available at markets and have high taste appeal [meaning those market products which have a pleasant taste as compared to wild ones].”

We documented the trade of 40% *(n* = 59) of the reported taxa in local markets, in which the highest number of taxa (36 taxa) were bought from the markets by PO. The special foraging of *Apteranthes tuberculata* is performed by AI in the month of February when the availability of the species is considered high in the mountain pastures. *Apteranthes tuberculata* is an important economic taxon that is foraged by Afghan refugees and brought to local markets for sale. One of the AI participants commented about the importance of this species:

“We don’t have time in February because we have only one month for the foraging of Pamanai (*Apteranthes tuberculata*) and we hike 3 h every day to reach the particular patches of the mountains where it is available and collect 5 to 6 kg each day which pays good money compared to any other occupation”.

We have observed that among the quoted taxa, only 15% *(n* = 22) of taxa were reported to be stored, the majority of which were wild food plants (i.e., *Quercus incana*, *Pinus gerardiana*, *Mentha longifolia*). The most used plant parts were seeds (36%) followed by fruits (27%).

We observed that both AI and AO reported taxa that could be imported from Afghanistan for personal use without the need for permission (i.e., *Alkanna tinctoria*, *Berberis lycium*, *Buxus wallichiana*). The interviews also highlighted that the new border policy recently devised by Pakistani authorities has drastically decreased the opportunity of acquiring plant materials from the homeland of Afghanistan as now only those Afghans holding a valid passport, visa, or *rahdari* (a card issued since 2015 to facilitate frequent cross-border movement) are allowed to enter Pakistan [10].

### 2.6. Commonalities and Differences among the Three Groups

There is a remarkable difference in the ethnobotany of PO compared with both AI and AO (Figure 2). The three groups share 41 taxa, while in total the refugee groups (AI and AO) share 75 taxa. The least similar groups in terms of shared plant taxa are AI and PO with only 44 taxa (Jaccard Similarity Index = 0.37). Both these groups also mentioned 16 taxa not shared with the other groups.

Of the 133 identified taxa, 25 were mentioned by more than 40% of the interviewees in each group (Figure 3). Among those, only three were shared by the three groups (*Zygophyllum, Withania*, and *Apteranthes*). Fifteen taxa were found exclusively among local Pathans who shared only five top used species with the other groups. Indeed, local Pathans’ Jaccard Similarity Indexes drop to 0.11 with AI and 0.19 with AO, while the two refugee groups maintain a high similarity index (0.50).

Local Pathans reported a maximum of 18 idiosyncratic plants, the majority of which were WFPs, while the two groups of refugees (AI and AO) reported a fewer number of wild taxa.

Nearly one-third of WFPs are commonly shared by the studied groups (Figure 4). Afghan Pathans living outside of camps reported the largest number (20) of idiosyncratic WFPs used, while AI reported a smaller number (9) of idiosyncratic WFPs. Local Pathans also reported 10 idiosyncratic WFPs. The most similar groups are the two refugee communities, yet their divergence from PO is less evident than for the overall taxa.

Sixteen WMPs and WVPs were commonly shared by the studied groups (Figure 5). The highest similarity index was found between the two refugee groups (JI = 0.44). Afghan Pathans living inside camps quoted the greatest number of medicinal and veterinary plants and reported the maximum of 23 idiosyncratic taxa among their overall use of 70 taxa.

In comparison with AI, and somehow paradoxically, AO reported fewer medicinal and veterinary plants despite the fact that this group also comprised Kochi Afghans (pastoral nomads) and practice pastoralism, and thus they are thought to have remained in close association with their natural environment and have extensive information about traditional remedies. Elderly community members still use traditional plants for the treatment of various diseases and the majority of the study participants from inside the camps kept a lot of medicinal plants in their houses. A 65-year-old elderly AO participant commented:

“In the past, we used the majority of medicinal plants, and we did not go to the hospital until some big problem happened to us, because we had a strong belief in those plants and that was the reason that we recovered soon after taking those folk remedies. But today we do not use all those plants because of the availability and use of the Western medical system and drugs. Now if we get a chance to use the traditional remedies against any disorder, we are not mentally satisfied with the results of traditional remedies despite them working”.

The above statement shows that AO had strong beliefs about traditional therapies in the past, but due to the interference from the Western medical system, they have nearly lost this connection.

## 3. Discussion

The results show two main findings. First, Afghan Pathans outside the camps tend to collect and use more wild food plants, as already shown in a previous study conducted by some of the current authors in the Mansehra area [8], most likely because their exposure to nature is higher than that of Afghan Pathans living in camps. However, refugees inside the camps paradoxically use more medicinal plants than Afghan Pathans living outside the camps, possibly because of more disadvantaged economic conditions, making it more difficult for them to access Pakistani health services. These two divergent variables (decreased exposure to the natural environment and difficulties in accessing health care services due to social marginalization) have shaped different trajectories with respect to the wild food and medicinal plants that they gather and use.

However, while Afghan ethnobotanies show some convergences, significant differences do exist with the ethnobotany of local Pathans, who use far fewer plants. Although Afghan refugees in fact utilize wild plants from across different habitats in both Afghanistan and Pakistan (also via family networks), this does not occur for local Pathans, who use wild plants found only in their environment.

### 3.1. Afghan Refugees Share Similar Ethnobotanical Knowledge, with Some Differences

More than half of the identified taxa were common to AI and AO, possibly due to the fact that both these communities inhabited the same socio-ecological environment before their migration and had similar cultures and customs. Additionally, the two groups are exogamous, which allows intermarriages and consequently promotes exchanges of LEK, mainly through vertical transmission (transferred from grandmother to mother and from mother to daughter); a similar finding has been documented in other studies [11,12].

Moreover, we found a remarkable divergence in the ethnobotany of PO with both AI and AO, possibly because local Pathans have lived together with other Pakistani ethnic groups for centuries, sharing the same religious faith (Sunni Muslim) and social life, while Afghans relocated only a few decades ago. According to some authors [13], the nature and degree of contact and exchange with neighboring socio-culture groups may determine the level of homogenization of LEK. Nearly one-third of the taxa reported by AI and AO were not quoted by PO, possibly due to the fact that the majority of the reported taxa were harvested in Afghanistan where refugees regularly visit their relatives, friends, and family. This may reinforce the vulnerability of refugees’ ethnobotanical knowledge as a result of a kind of “displaced” foraging (living in one country and foraging in another one). However, the strong kinship ties that Afghan refugees in Pakistan have always fostered with their relatives in their home country could also allow them to easily acquire dried medicinal plants without any actual gathering.

The comparatively high number of idiosyncratic plants reported by local Pathans is possibly due to their centuries-old familiarity with the natural environment. Additionally, they are highly dependent upon farming which could enhance the resilience of LEK related to plant resources. We observed that both refugee groups are economically disadvantaged and sometimes tend to work in cities, leaving little time for farming and foraging. One of the AI study participants commented:

“The economic condition of the majority of the families is not good, so from a very small age, we try to engage our children in different kinds of work, e.g., working as a laborer or trying to run a small shop in the markets or in most cases we send them abroad to earn money. So after returning, we neither want to discuss wild plants nor have time for foraging”.

The above statement suggests that the progressive weakening of the plant–people relationship between the older and younger generations could probably further lead to the erosion of LEK.

While the higher consumption of wild food plants among Afghan Pathans living outside the camps is possibly due to their greater opportunity of roaming in the environment as a result of their agricultural activities, the smaller number of plants mentioned by the refugees living within camps could be due to their more robust food security, as food is often provided to them by humanitarian organizations.

On the other hand, the higher number of wild medicinal plants (both in human and veterinary medicine) among AI (vs. AO) may be due to two factors: (a) refugees inside the camps have more limited access to proper health services, and therefore they have to rely more on their traditional plant medicine; (b) camps in the study area include many Kochi Afghans (traditional pastoral nomads), who still retain vast ethnoveterinary knowledge.

### 3.2. Refugees Utilize Wild Plants across Habitats and Borders

Nearly two out of five plants mentioned in this study are foraged in Afghanistan, only one out of five in Pakistan, and two out of five in both countries.

Local Pathans tend to forage plants in the fields near their houses, while refugees outside the camps foraged until recently in the mountains in Afghanistan. Indeed, the proximity of the plants to villages could possibly lead to an increase in the knowledge and use of synanthropic plants [14,15,16], while they may have less knowledge of those plants which grow far from their houses [17,18,19].

Until the recent past, Afghan refugees (outside the camps) used to continually visit relatives in their homeland, and, as described in other studies [20], these family networks allow diasporic groups to acquire plants transnationally: when people go to visit friends and families in their place of origin, dried plant materials are acquired before traveling back to the host country [21]. In recent years, however, border rules between Pakistan and Afghanistan have become stricter and our study participants agreed that these trans-border movements of people and plants are more difficult now.

Additionally, many informants feel uneasy about foraging in the mountains in their host country (Pakistan). One of the AI participants commented:

“Without an expert or a local person, it is an unwise decision to go for research or the collection of wild plants because something bad could happen to us”.

The above statement shows that the refugees do not feel comfortable in their new environment, whereas, despite the difficulties in crossing the border and the increasing security problems in Afghanistan, they still wish to go forage in their homeland. A 71-year-old AI mentioned:

“In Afghanistan, when we go to visit the mountains, people show great interest in collecting wild cumin while returning to their own homes as it is used as a flavoring agent in our food”.

The AO group reported a high number of idiosyncratic uses of WFPs, which might possibly be associated with the fact that the majority of the quoted taxa derives from their homeland, where they regularly visit their relatives and place of origin in Afghanistan. In addition, the community has daily exposure to nature and practical activities which may lead to enhancing the resilience of LEK. The divergences in LEK regarding WFPs of Afghan Pathans outside the refugee camps are probably linked with their original home places, which are sometimes geographically isolated, as other works have pointed out (for example [6]) isolation could enhance the resilience of robust LEK.

On the other hand, possible reasons for the abandonment of foraging practices by Afghan Pathans inside the camps could be that they mainly live on remittances sent from abroad and they are mostly dependent on the cultivated vegetables available in markets. One of the AI study participants commented:

“The camp environment is not suitable for finding a large number of wild food plants, so for this purpose, we would need to go to the mountains, which I think is hard work and a time-consuming process—that is the reason we go for those products which are easily accessible and available in the markets”.

The continuation of plant use may also be related to the legal aspects of cross-border plant imports. We observed that both AI and AO reported plant taxa (i.e., *Alkanna tinctoria*, *Berberis lycium*, *Buxus wallichiana*) widely imported from Afghanistan for personal use without the need for permission. Our findings are in line with the work of [22] in the Netherlands, where Surinamese migrants in Amsterdam heavily relied on importation because of the flexible Dutch entry laws. Conversely, the strict entry law for plant imports into the UK makes it difficult for Latino migrants to acquire some of their homeland plants [4,23].

Since 2001, however, due to the unrestricted movement of militants across the Pakistan–Afghanistan border, both countries have faced internal security threats [24], and so to improve its control over the border, Pakistan has developed certain strategies such as closing or fencing the border and tightening rules at the border crossing for Afghans [25]. These changes in border policies have ultimately weakened the bond tying Afghan refugees to their motherland and will heavily affect the import of plant materials from Afghanistan and therefore the erosion of LEK of the refugees; similar findings were documented among Afghan refugees living in Mansehra [8]. One AO study participant mentioned:

“Due to the new border policies and strictness, we are not even able to meet with our family members who remain in Afghanistan nor can they come to meet us because financially it is very costly and difficult to continue visiting our motherland. We have never experienced such restrictions since the time of migration until now and could have visited whenever we wanted”.

We also observed that due to new border policies Afghan pastoralist communities have abandoned pastoralism because in summer they cannot move their herds into the mountain pastures in Afghanistan, since border laws are no longer flexible. According to one AO participant:

“It is quite hard for us to keep our animals in a very hot climate in summer instead of moving them into the mountain pastures when cross-border movement is stopped and we face some major problems like the scarcity of water and fodder for animals. That’s why we have now sold our herds and are trying to invest the money in different businesses”.

We also observed in our study that Kochis Afghan inside the refugee camps sometimes rent a piece of land in the Pakistani mountains from local PO upon agreement; a few family members are therefore sent in the summer to the pastures with their herds while the rest of the family remain in the camp. The majority of the ethnoveterinary plant uses are not actively practiced by the study participants anymore but are only remembered. One of the AI study participants (a 53-year-old man) commented:

“We don’t want to keep animals in our houses because of certain reasons: we are given a very limited place to live where the whole family can hardly survive, the structure and texture of the soil inside the camp do not look suitable for growing fodder for animals, and the last is the poor economic position of the local communities”.

Indeed, due to poor economic status and government restrictions, the majority of the population inside the camps do not keep any domestic animals in their houses, and only a few families have reported keeping one or two domestic animals in their houses. Moreover, the high number of plants mentioned by Afghan refugees may be misleading because the plants are mentioned by very few interviewees.

Recent developments in the political landscape of Afghanistan were not specifically considered in our field study. However, the overthrow of the Ashraf Ghani government on 15th August 2021 has further drastically affected the lives of Afghans in Pakistan as well. Many of the diasporas do not want to visit Afghanistan in the current situation, as its economy is collapsing. As the bank accounts of millions of citizens in Afghanistan have been frozen and most people have become jobless, many Afghan citizens are selling their possessions to buy some food items and face food insecurity even in urban areas. All these factors have led the diaspora in Pakistan to discontinue regular visits to their homeland, families, and friends, which ultimately has made the borders even tougher than our study could highlight.

## 4. Materials and Methods

### 4.1. Study Area and Communities

The field survey was conducted in Kohat District (1,112,452 inhabitants: [26], NW Pakistan. Most of the population lives in rural areas (75.71%). The average elevation of Kohat District is 489 m.a.s.l., and it is located between 70°34′ and 72°17′ E and 32°47′ and 33°53′ N (Figure 6). 

The area comprises a succession of irregular mountains, ranging from 610 to 1526 m which are separated by open valleys. In high mountainous areas winter is cold (−5 °C to 3 °C) while summer remains fresh (10 °C to 15 °C), in plain areas the summer is extremely hot with temperatures reaching 50 °C, while the winter remains mild. Due to the low precipitation rate, there are no running streams in the hills and water for agricultural purposes is scarce, as the soil is mostly sandy and stony (Figure 7). The vegetation consists of several xerophyte plants, such as *Monotheca buxifolia*, *Withania coagulans*, *Senegalia modesta*, *Peganum harmala*, and *Calotropis procera*. In plain areas, people sometimes are able to grow guava in gardens for economic purposes. Kohat has also a well-developed hydrocarbon extraction and mining. The area is home to various groups of people associated with different spoken languages and various religious faiths. Indeed, in the early sixteenth century three main Pathan tribes, namely Afridi, Banoori, and Bangash, settled in Kohat District, which remained a part of the Mughal Empire until the eighteenth century [27].

In 1978, after the takeover by the communist party in Afghanistan, Afghan citizens started migrating to neighboring countries such as Pakistan and Iran. As many as 193,000 Afghan refugees received asylum in Pakistan by the end of 1979 [28]. A significant acceleration in the influx of refugees occurred after the invasion of Afghanistan by Soviet forces when 80,000 to 90,000 Afghan Pathans crossed the Pakistani border every month between January and December 1980 [29]. In 1989, the number of Afghan refugees (3,270,000) reached its peak, which represented 3% of the total population of Pakistan at that time [30,31].

In 1992, over a period of six months, approximately 1.2 million Afghan refugees moved back to their own country from Pakistan after the fall of Kabul to Mujahidin, and by the start of 1994 the population had fallen from 3.2 million to 1.47 million. However, the migration of refugee communities back to Afghanistan occurred for only a short time because of the continuation of the civil war [32]. In late 2001, when the Taliban regime came to an end, the government of Pakistan devised a plan to send all Afghans back to Afghanistan. In 2002, an agreement was signed between united nations high commissioner for refugees (the Government of Afghanistan, and the Government of Pakistan, according to which all returns had to be performed willingly. However, in the summer of 2005, the Government of Pakistan started to close different refugee camps in different tribal agencies, such as North and South Waziristan, Bajaur, and Kurram, which resulted in the displacement of approximately 200,000 Afghan refugees, who then returned to Afghanistan.

The Government of Pakistan constructed several dozen camps in the North-West Frontier Province that remained safe from the most critical problems (such as malnutrition and epidemics) that a largely displaced population face. Since the Soviets invaded Afghanistan, the Government of Pakistan has given basic rights to and has adopted a liberal policy toward the Afghan refugees. According to the 1973 constitution, only Pakistani citizens have a right to stay in their choice of residence and freedom of movement, but the Government of Pakistan grants the same privileges (i.e., to reside where one chooses, travel throughout the country, do business) to all registered Afghans.

In 2011, only 33% of the population was reported as still residing inside the camps, while the remaining 67% had moved into various urban and rural areas [33]. NW Pakistan hosts the majority of the Afghan diaspora (58.1%) in the country [34].

### 4.2. Data Collection

The field ethnobotanical study was conducted over a period of five months from February to June 2022. Ninety study participants were conveniently selected among middle-aged and elderly community members, later adopting a snowball sampling technique. Convenience selection is a type of nonprobability or nonrandom sampling “where members of the target population that meet certain practical criteria, such as easy accessibility, geographical proximity, availability at a given time, or the willingness to participate are included for the purpose of the study” [35]. The interviewees, 30 individuals from each studied group, were mainly elderly adults ranging from 50 to 85 years of age because they were thought to be more knowledgeable in terms of LEK (Table 3). The study participants belonged to three groups, i.e., Afghan Pathans inside the refugee camps (AI), Afghan Pathans outside the refugee camps (AO), and local Pathans, naturally living outside the refugee camps (PO).

Afghan Pathans living outside the camps perform two types of pastoralism: one in which they herd their animals near their houses, and a second in which they rent pastures from local Pathans and send 2–3 family members with their herds into the mountain pastures, while the rest of the family remains in their homes. Afghan Pathans have little opportunity of sharing knowledge with local Pathans as they are not in close contact, despite the fact that they live in the same environment. Refugee groups have developed a certain social network to acquire some medicinal plant remedies in Kohat city, Kurram Agency (an area sharing a border with Kohat District), and from their homeland whenever they travel there to visit their friends and families.

All the interviews were conducted by the first author in the local language, Pashto, which was the first language of all the study participants.

Before commencing each interview, prior informed consent (PIC) was obtained verbally, and the *Code of Ethics* of the International Society of Ethnobiology (ISE) [36] was strictly followed. Information about folk wild plant use was obtained through free listing first, and semi-structured interviews afterward. The questions mostly focused on the medicinal, veterinary, and food uses of wild plants, in the present and in the past (e.g., childhood); additionally, some culturally relevant uses of wild plants as fuel and cosmetics/fragrances were recorded. For each botanical taxon, the researcher asked about the exact homemade preparations and uses. All available plants were collected while in the field, photographed, dried, and identified with the help of a plant taxonomist at the Department of Botany, University of Peshawar. Voucher specimens of all documented plants were prepared and subsequently deposited in the Herbarium of the Department of Botany at the University of Peshawar. Those taxa for which no specimen could be collected were identified via folk plant names, photographs, and detailed ecological and plant descriptions. Moreover, study participants gave their consent to publish the photos that they had taken on their own. The nomenclature followed World Flora Online [37] for plant taxa and Index Fungorum [38] for mushroom taxa, with family assignments aligned with the current Angiosperm Phylogeny Group IV recommendations [39].

This study has as a main limitation an unavoidable sampling bias with regard to gender. Despite the fact that women retain a large part of the folk plant knowledge, in this study, only a few women were interviewed. This is because in Pathan/Afghan cultures it is often inappropriate or undesirable for men to talk to women due to the extremely conservative attitude regarding social relations in the area from both a cultural (observing the strict Pashtunwali or Pakhtunwali code of conduct) and religious (within a rigid Sunni Muslim sphere) point of view. Unfortunately, it was impossible for our research group to involve Pathan female scientists, due to the fact that their families did not allow them to spend days and nights with male colleagues in the field. Nevertheless, we were able to interview a few elderly women outside of their houses when they were willing to talk.

### 4.3. Data Analysis

The documented wild plant ingredients were organized in MS Excel. We utilized emic food categories as much as possible. Raw snacks referred to (parts of) plants eaten on the spot without any processing. According to this definition, salads are not considered raw snacks, as they are eaten after some preparation and during meals. The data were categorized into four datasets: one containing all identified taxa, the second one comprised of the identified most frequently quoted taxa (reported by more than 40% of the informants), the third one containing wild food plant taxa, and the fourth one comprised medicinal and veterinary taxa. These four datasets, which were generated for each of the three considered groups, were compared through proportional Venn diagrams and the Jaccard Similarity Index (for each pair of datasets) following the application designed for the use of this ecological index in the ethnobotanical domain [40].
$$Jaccard\;Index=\frac{Number\;of\;plant\;taxa\;reported\;by\;both\;group\;A\;and\;group\;B}{Total\;number\;of\;plant\;taxa\;reported\;by\;group\;A}$$

## 5. Conclusions

The current research yielded two main findings. Firstly, there is a general convergence of the ethnobotanies of Afghan refugees living inside and outside the camps. This is possibly due to the fact that both refugee communities used to live in the same environment before relocating and have the same culture and customs.

However, the study shows some subtle differences between the refugees living outside and inside the camps: the former use more wild food plants, while the latter uses more medicinal plants.

This can be explained by two concomitant phenomena: (a) Afghan Pathans outside the camps still have extensive exposure to the natural environment, which allows them to forage more frequently and consistently; (b) Afghan Pathans inside the camps have limited access to health services—because of their disadvantaged economic conditions—and therefore rely more on traditional plant remedies for managing their health.

Secondly, the data show significant differences between Afghan and Pathan ethnobotanies, with the refugees knowing more about wild plant uses than local Pathans. In fact, until the recent past, refugees used to collect wild plants across different habitats in both Afghanistan and Pakistan, while this has never been the case for local Pathans. However, these Afghan transnational plants and human mobilities are fading, due to stricter border rules.

The main findings of the study suggest therefore that stakeholders should pay careful attention to facilitating (i) the proper settling of refugees, their engagement in farming, and thus increased exposure to the natural environment and plant foraging practices; and (ii) the transnational movement of refugees since this could facilitate the further resilience of their traditional medical plant systems; both of these measures could ultimately enhance refugees’ food security, health status, and well-being.

## Figures and Tables

**Figure 1 plants-12-00574-f001:**
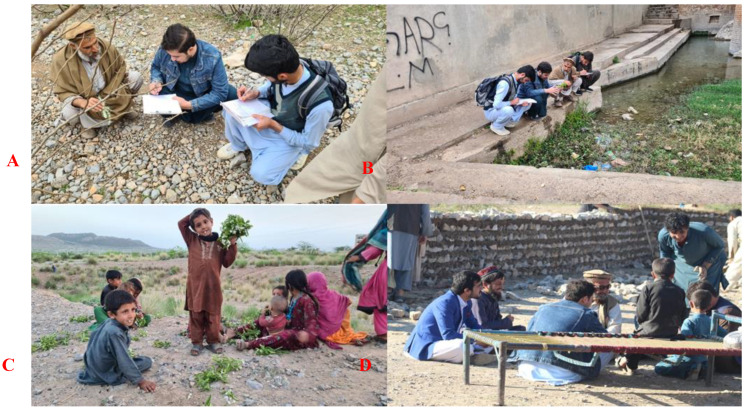
(**A**) The first author with their research assistant interviewing informants about *Cirsium arvense*. (**B**) The first author with their research assistant collecting specimens of *Adiantum capillus-veneris* outside a camp. (**C**) Children collecting the flowers of *Justicia adhatoda* outside a camp to eat the nectar. (**D**) The research team interviewing Afghan Pathans living in a refugee camp. The study participants gave consent to publish the photos.

**Figure 2 plants-12-00574-f002:**
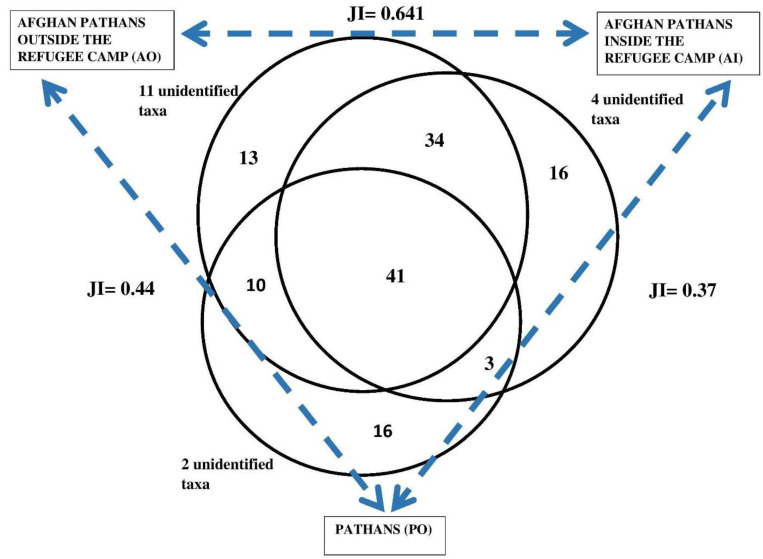
Venn diagram with the total identified quoted taxa and relative Jaccard Similarity Indexes (JI). We recorded 16 unidentified taxa, but 1 unidentified species was mentioned by AI and AO.

**Figure 3 plants-12-00574-f003:**
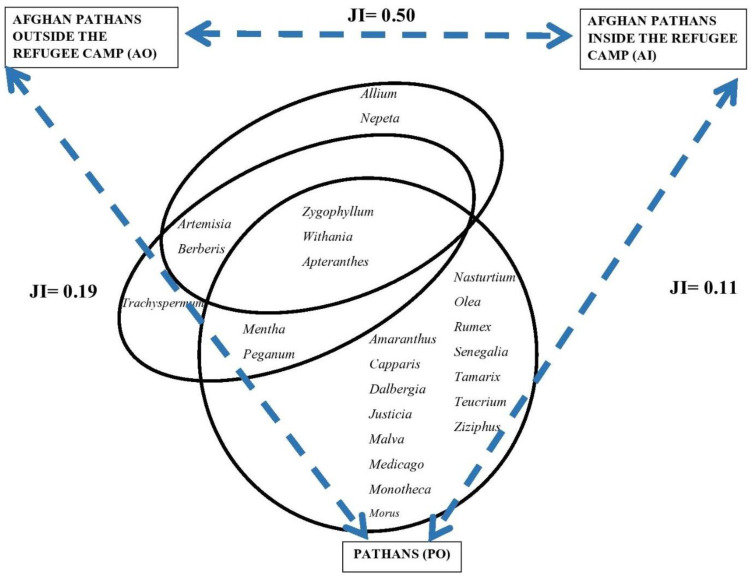
Venn diagram for the most frequently reported taxa (more than 40% of the participants in each of the studied groups) and Jaccard Similarity Indexes (JI).

**Figure 4 plants-12-00574-f004:**
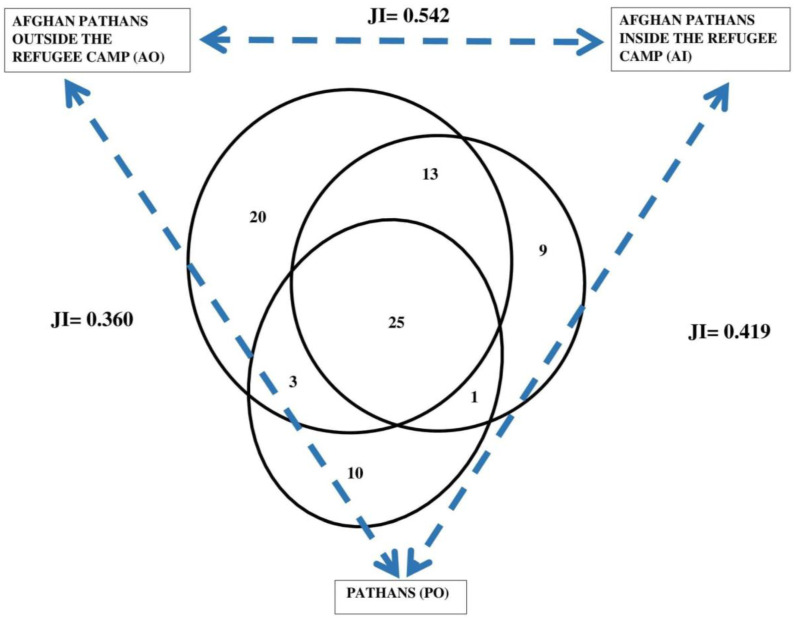
Venn diagram for the WFP taxa reported by the studied groups and Jaccard Similarity Indexes (JI).

**Figure 5 plants-12-00574-f005:**
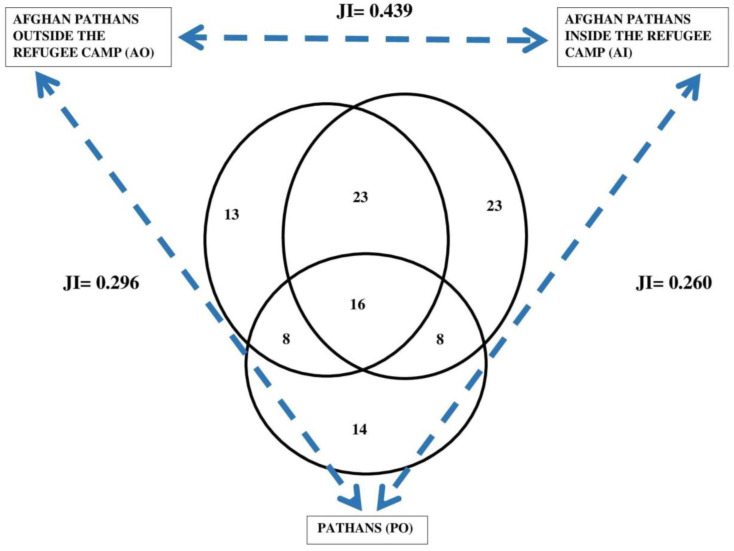
Venn diagram for the medicinal and veterinary plant taxa reported by the studied groups and Jaccard Similarity Indexes (JI).

**Figure 6 plants-12-00574-f006:**
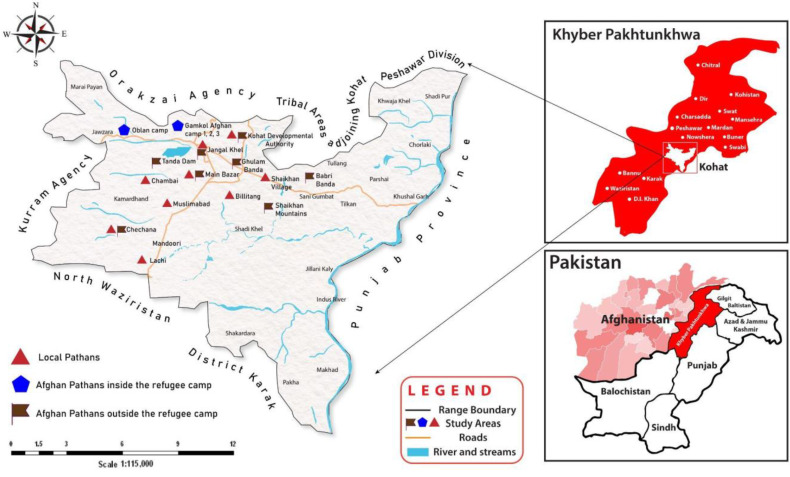
Map of the study area (Credit: Sheheryar Khan).

**Figure 7 plants-12-00574-f007:**
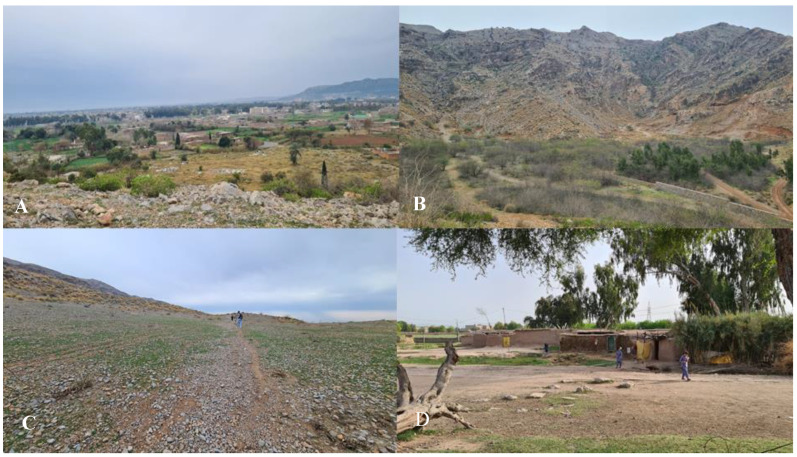
(**A**) The landscape of an area where local Pathans live. (**B**) The mountains where refugees herd their animals. (**C**) The landscape of the area around Gamkol Afghan Camp. (**D**) Inside view of Oblan refugee camp.

**Table 1 plants-12-00574-t001:** Wild plants recorded among the three studied communities.

Plant Taxon; Botanical Family; Voucher Specimen Code	Local Name (in Pashto)	Part Used	Procurement	Collection Site	Ethnobotanical Uses and Their Recipes	Frequency of Quotation		
						AI	AO	PO
*Achillea santolinoides* Lag, Asteraceae #	Zawel ^AI, AO^	Flowers ^AI, AO^ Fruits ^AI^ Leaves ^AI^	FA ^AI. AO^ BM ^AI^	Fields, field edges, and mountains	M: flowers and fruits, heated on a metal plate (*Teghna*) and ingested, or sometimes boiled and ingested to treat chest pain and dysgeusia, and as anti-cough and antipyretic agents; infusion (fresh flowers) to treat gastric gas and acidity V: leaves, fodder for goats O: leaves, for fuel	++	++	-
*Achyranthes aspera* L., Amaranthaceae, UOP NK 00101	Pothkanda ^PO^ Damia ^PO^	Flowers ^PO^	FP ^PO^ BM ^PO^	Fields and stream sides	M: powdered, mixed with one teaspoon of honey, anti-cough	-	-	+
*Adiantum capillus-veneris* L., Pteridaceae, UOP NK 00102	Pathosnara ^AI, AO^	Whole plants ^AI, AO^	FA ^AI, AO^ FP ^AI, AO^	Stream and field edges	M: decoction, to treat measles, and intestinal pain and spasms, and as an antipyretic and anti-diarrheal in children; or sometimes decoction with mint, as an anti-diarrheal; filtrate, topically applied, anti-itching	++ *	++	-
*Agaricus* spp., Agaricaceae #	Dangarkai ^AO^ Samarokh ^AO^ Gugushtak ^AO^	Aerial parts ^AO^	FA ^AO^ BR ^AO^ FP ^AO^	Mountains	F: fried in oil with other greens M: intestinal spasms	-	+	-
*Alhagi maurorum* Medik., Fabaceae #	Zuz ^AO^	Leaves ^AO^ Flowers ^AO^	FA ^AO^ FP ^AO^	Fields	M: infusion to lower body temperature	-	+	-
*Alkanna tinctoria* Tausch, Boraginaceae *#*	Ghuralang ^AI, AO^	Root bark ^AI^	BR ^AI, AO^ BM ^AO^ FA ^AI, AO^	Mountains	M: powdered, mixed with *Brassica* oil, topically applied to heal burns, or sometimes powdered and mixed with oil (*dasi ghee*), drunk to heal internal wounds	+	+	-
*Allium carolinianum Redouté*, Amaryllidaceae, UOP NK 00103	Piazakea ^AI^, Ghra piazakea ^AO^	Bulb ^AI, AO^ Leaves ^AI, AO^	FP ^AI, AO^ FA ^AO^	Mountains	F: bulb and leaves, as raw snacks, or sometimes fried in oil with tomatoes, chilies, and other greens, also used in different sauces	+	+	-
*Allium rosenbachianum* Regel, Amaryllidaceae *#*	Arghankea ^AI^ Khezea ^AI, AO, PO^	Leaves ^AO, PO, AI^ Roots ^AO, AI^	FA ^AI, AO^ BM ^AO, AI, PO^	Mountains	F: leaves and roots, as raw snacks *, or sometimes boiled, then fried in oil with tomatoes, chilies, and other greens O: root extract, as glue	+++ *	++	+
*Allium* sp., Amaryllidaceae #	Chamangchan ^AI^	Whole plants ^AI^	FA ^AI^	Fields and mountains	F: fried in oil	+	-	-
*Aloe vera* (L.) Burm.f., Asphodelaceae, UOP NK 00104	Karghandal ^AI, AO, PO^ Zargera ^AO^ Zarpanra ^PO^	Gel ^AI, AO, PO^	FA ^AI^ FP ^AI, AO, PO^	Graveyards and mountains	F: fried, in oil (*dasi ghee*) along with meat (*keema*) M: remedy against joint pain, or sometimes eaten with salt to lower stomach acidity, glucose level, and cholesterol level, or topically applied for beautiful skin, for anti-hair loss, and to treat external wounds, acne, and melasma	+	++	++
*Amaranthus viridis* L., Amaranthaceae, UOP NK 00105	Sarkhmal ^PO^ Ranzaka ^AI, AO, PO^	Aerial parts ^AO, PO, AI^	FA ^AI, AO^ FP ^AO, AI, PO^	Maize and other fields, roads, and stream sides	F: boiled and then fried in oil with tomatoes, chilies, and other greens	++	++	+++
*Apteranthes tuberculata* (N.E.Br.) Meve and Liede, Apocynaceae, UOP NK 00106	Pamanai ^AI, AO^ Pamankai ^AI^ Chong ^PO^	Aerial parts ^AI, AO, PO^	FA ^AI, AO^ FP ^AI, AO, PO^ BM ^AI, AO, PO^	Mountains	F: consumed as a salad, also fried in oil with meat and sometimes with other greens M: filtrate, anti-diabetic, hypertension, flatulence, and blood purification	+++	+++	+++
*Artemisia absinthium* L., Asteraceae #	Marvai ^AI^	Seeds ^AI^ Shoot tips ^AI^	FA ^AI^	Foothills, mountains, and non-cultivated land	M: seeds, chewed to treat flu and gastrointestinal tract pain; shoot tips are boiled in water to treat flu in children	+	-	-
*Artemisia* sp., Asteraceae, #	Tarkha ^AI, AO^	Whole plants ^AI, AO^	BR ^AI, AO^ FA ^AI, AO^ FP ^AI, AO^	Fields and mountains	M: cold macerate (let macerate for one night under the sky/stars and use before breakfast) to treat hepatitis, stomach problems, diarrhea, fever, and to remove intestinal worms *; infusion (leaves), anti-diabetic and antipyretic O: leaves, fuel	+++ *	+++	-
*Asparagus officinalis* L., Asparagaceae, UOP NK 00107	Lakhtey ^AI, AO, PO^ Marchoghnrhi ^AO, PI^ Khezee ^PO^	Young shoots ^AI, AO, PO^	FA ^AI^ FP ^AI, AO, PO^ BM ^AI, AO, PO^	Fields, foothills, and mountains	F: boiled and then fried in oil with meat (*keema*) M: remedy is used against diabetes	+	++	++
*Atriplex laciniata* L., Amaranthaceae, UOP NK 00108	Shorakai ^PO^	Whole plants ^PO^	FP ^PO^	Fields	F: leaves, consumed as a salad, also cooked with other greens V: leaves, as fodder O: whole plant, fuel	-	-	++
*Avena sativa* L., Poaceae, UOP NK 00109	Zawdar ^PO^	Whole plants ^PO^	FP ^PO^	Crops, fields	V: fodder, also increases meat O: acts as a galactagogue	-	-	+
*Bauhinia variegata* L., Fabaceae, UOP NK 00110	Kachnar ^AO, PO^ Khezya ^PO^	Young flowers ^AO, PO^	FA ^AO^ FP ^AO, PO^ BM ^PO^	Fields	F: fried in oil along with other greens	-	+	++
*Berberis lycium* Royle, Berberidaceae, UOP NK 00111	Ziar largai ^AI, AO, PO^ Kurai ^AI, AO^ Zark ^PO^	Fruits ^PO^ Roots ^AI, AO, PO^	FA ^AI, AO^ BR ^AI, AO^ BM ^AI, AO, PO^	Fields and mountains	F: fruits, as raw snacks M: roots are powdered and mixed with milk and egg yolks to treat back pain, bone fractures, joint pain, cough, and tuberculosis phlegm, or sometimes the powder is mixed with black tea and honey to treat tuberculosis *, or sometimes the powder is sprinkled over a half-boiled egg to treat chest pain, kidney wounds, and flu; decoction to treat chest tightness in children V: decoction to heal internal and external wounds	+++ *	+++	+
*Buxus wallichiana* Baill., Buxaceae #	Shamshad ^AI^	Leaves ^AI^	BR ^AI^ BM ^AI^	Mountains	M: infusion, anti-diabetic	+	-	-
*Calotropis procera* (Aiton) W.T.Aiton, Apocynaceae, UOP NK 00112	Spalmai ^AI, AO, PO^	Leaves ^AI, AO, PO^ Stem juice ^AO^ (madar juice)	FA ^AO^ FP ^AI, PO^	Fields and foothills	M: leaves, powdered and then topically applied over skin and snake bite areas to remove thorns, and to treat seborrheic keratosis, as an anti-diabetic, and in antidotes, or sometimes leaves are punctured with a needle and placed in shoes, anti-diabetic; madar juice, topically applied to stop bleeding V: leaves as fodder, kills intestinal worms, sometimes the insects (*Poekilocerus pictus*), that feed on this plant are caught and powdered, which is then put in the nose of animals to kill brain worms; madar juice, topically applied over the skin and mammary glands of cows to remove piles and ticks O: madar juice, used in fermentation	+	++	++
*Capparis decidua* (Forssk.) Edgew., Capparaceae, UOP NK 00113	Kerrha ^AI, AO, PO^	Fruits ^AI, AO, PO^ Stems ^PO^	FA ^AI^ FP ^AO, PO^ BM ^AI^	Fields and mountains	F: fruits, as raw snacks M: decoction, anti-diabetic V: stems, as fodder, anti-bloating	+	++	+++
*Caroxylon imbricatum* (Forssk.) Moq., Amaranthaceae, UOP NK 00114	Zmea ^PO^	Leaves ^PO^	FP ^PO^	Foothills and mountains	O: leaves, as soap *	-	-	++ *
*Carthamus oxyacanthus* M.Bieb., Asteraceae, UOP NK 00115	Konzala ^PO^	Seeds ^PO^	FP ^PO^	Fields	F: seeds, used in various confections such as patasha M: seeds, mixed with jaggery to treat urethritis and to remove waste material from the kidneys and stomach	-	-	++
*Carum carvi* L., Apiaceae, UOP NK 00116	Zanrkai ^AI, AO, PO^	Fruits ^AI, AO, PO^	FA ^AO^ FP ^AO, PO^ BM ^AI, AO, PO^	Mountains	O: seeds, as a spice	+	++	++
*Cassia fistula* L., Fabaceae, UOP NK 00117	Chumberhyal ^AI, AO^ Tur largee ^AO^ Maltas ^PO^	Seeds ^AO, PO^ Fruits ^AI, AO^	FA ^AI^ FP ^AI, AO, PO^ BM ^AO, PO^	Home gardens, fields, and foothills	M: seeds, powdered and mixed with water to treat constipation and stomach pain in children; decoction (seeds), used as a gargle for throat infections, or sometimes topically applied as anti-hair loss; fruits, mixed with mother milk to treat fecal infections in children	++	++	+
*Cedrus deodara* (Roxb. ex D.Don) G.Don, Pinaceae, #	Almanza ^AI, AO^ Deyor ^AI^	Seeds ^AO^ Gum ^A^	BR ^AI, AO^ FA ^AI, AO^	Mountains	F: seeds, as raw snacks for relaxation and to release tension M: gum, boiled in water and drunk to heal wounds	+	+	-
*Chenopodium album* L., Amaranthaceae, UOP NK 00118	Surmi ^AO, PO^ Bato sag ^AI, AO, PO^	Aerial parts ^AI, AO, PO^	FA ^AI, AO^ FP ^AI, AO, PO^	Fields (especially maize fields)	F: boiled and then fried in oil with tomatoes, chilies, and other greens M: remedy to treat gastrointestinal tract problems	+	++	++
*Cichorium intybus* L., Asteraceae #	Shin Gulayi ^AI^ Tariza ^AI^ Shamakea ^AI, AO^	Roots ^AI, AO^	FA ^AI, AO^ FP ^AI, AO^	Wheat fields	M: roots, powdered and mixed with water to treat headaches and shoulder pain, and as an antipyretic *	+ *	++	-
*Cirsium arvense* (L.) Scop., Asteraceae, UOP NK 00119	Spin azghayi ^AI^	Leaves ^AI^ Fruits ^AI^	FP ^AI^	Fields and mountains	V: leaves, as fodder; fruits, as fodder for various birds O: acts as a galactagogue *	+ *	-	-
*Citrullus colocynthis* (L.) Schrad., Cucurbitaceae, UOP NK 00120	Maraghuney ^AI, AO, PO^	Seeds ^AI, AO, PO^ Fruits ^AO^	FA ^AI^ FP ^AO, PO^ BM ^PO^	Mountains and sandy areas	M: seeds, 2–3 pieces are taken orally with water to treat stomach pain, gas, diabetes, and stomach worm, or sometimes powdered and mixed with water to make a solution, in which the legs are dipped as an anti-diabetic, or sometimes fruits and seeds are mixed with jaggery to form a jam to treat stomach problems V: seeds, boiled to treat appendicitis in cows and donkeys	++	++	++
*Citrus medica* L., Rutaceae #	Kambar ^AO^	Stem ^AO^	BR ^AO^	Fields	F: fried in oil	-	+	-
*Convolvulus arvensis* L., Convolvulaceae, UOP NK 00121	Parwatyea ^AI^	Leaves ^AI^	FA ^AI^ FP ^AI^	Fields	F: boiled and then fried in oil with other greens M: remedy to soften the intestine, but excessive use may cause diarrhea	+	-	-
*Cucumis melo* L., Cucurbitaceae, UOP NK 00122	Karkondai ^PO^	Fruits ^PO^	FP ^PO^	Fields	F: fruits, as raw snacks	-	-	+
*Cuminum cyminum* L., Apiaceae #, and possibly *Elwendia persica* (Boiss.) Pimenov and Kljuykov, Apiaceae #	Zera bute ^AO^	Fruits ^AO^	BM ^AO^	Mountains	O: as a spice	+	-	-
*Cupressus sempervirens* L., Cupressaceae, UOP NK 00123	Sabarghota ^AO^ Sabardana ^PO^	Seeds ^AO, PO^	FA ^AO^ FP ^PO^	Fields and mountains	M: seed extract, filtered, used to brush teeth to treat toothaches, or sometimes topically applied over the mammary glands to increase milk flow in mothers	-	+	+
*Cydonia oblonga* Mill, Rosaceae #	Boyo ^AO^ Boyee ^AO^	Fruits ^AO^	FA ^AO^ BR ^AO^	Fields	F: fruits, as raw snacks	-	+	-
*Cynodon dactylon* (L.) Pers., Poaceae, UOP NK 00124	Barawa ^AI^	Aerial parts ^AI^	FA ^AI^ FP ^AI^	Fields and mountains	M: chewed, the extract is topically applied to heal external wounds and to stop bleeding *, or sometimes boiled and then mixed with goat milk as an appetite suppressant in children *	+ *	-	-
*Dalbergia sissoo* Roxb. ex DC., Fabaceae, UOP NK 00125	Shawa ^AO, PO^	Leaves ^AO, PO^	FA ^AO^ FP ^AO, PO^	Fields	M: paste, topically applied on the head and feet for cooling the body, or sometimes fresh leaves are dropped in bath water as a cooling agent V: filtrate, used to treat foot and mouth disease and as an anti-bloating agent O: leaves, fuel	-	+	+++
*Descurainia sophia* (L.) Webb ex Prantl., Brassicaceae #	Khaksher ^AO, AI^ Jinjarak ^AO, AI^	Seeds ^AO, AI^	FA ^AO, AI^ BM ^AO, AI^	Fields and mountain	M: boiled and mixed with a teaspoon of honey and then eaten, or sometimes roasted on metal plate then powdered and mixed with milk, drunk to treat flu, cough, and chest pain, and as an antipyretic in children; decoction with sugar added as an appetite suppressant in children whose mothers suffer from hypo-galactorrhea	++	++	-
*Digera muricata* (L.) Mart., Amaranthaceae, UOP NK 00126	Soorgulai ^AI, AO, PO^ Katori sag ^PO^ Zangli palak ^PO^ Tandola ^PO^	Leaves ^AI, AO, PO^	FA ^AO^ FP ^AI, AO, PO^ BM ^PO^	Fields and mountains	F: boiled and then fried in oil, with onions and other greens	++	++	++
*Dodonaea viscosa* Jacq., Sapindaceae, UOP NK 00127	Zerawonea ^PO^	Whole plants ^PO^	FP ^PO^	Fields and mountains	O: as fuel wood	-	-	+
*Elaeagnus angustifolia* L., Elaeagnaceae, #	Sinzella ^AI, AO^	Fruits ^AI, AO^	BR ^AI, AO^ BM ^AO^ FA ^AO^	Fields and mountains	F: fruits, as raw snacks M: decoction, as an antipyretic and anti-cough agent	+	++	-
*Ephedra intermedia* Schrenk and C.A.Mey., Ephedraceae #	Uman ^AI, AO^	Leaves ^AO^ Young shoots ^AI^ Gum ^AO^	FA ^AI, AO^ BR ^AI, AO^ BM ^AI, AO^	Mountains	M: infusion to treat stomach pain, improves digestion, heals intestinal and stomach wounds *, or sometimes dried leaves are taken with water to treat stomach burning and as an antipyretic; decoction, applied externally on wounds developed during ear piercing; gum, topically applied to heal wounds	++ *	++	-
*Ferula assa-foetida* L., Apiaceae #	Barband ^AO^ Gandanar ^AO^	Stem latex ^AO^ Roots ^AO^	FA ^AO^ BM ^AO^	Mountains	M: stem latex, topically applied to heal wounds O: roots burned, and the smoke is used as anti-evil agent	-	+	-
*Ferula foetida* (Bunge) Regel, Apiaceae #	Hinja ^AO^	Whole plants ^AO^	FA ^AO^ BR ^AO^	Mountains	M: gum or stem latex, eaten as an anthelmintic V: as fodder O: as a spice, also acts as a galactagogue	-	++	-
*Ficus palmata* Forssk., Moraceae, UOP NK 00128	Inzar ^AO^ Inzarkayi ^PO^	Fruits ^AO, PO^	FA ^AO^ FP ^AO, PO^	Mountains	F: as raw snacks M: fruit extract, topically applied to remove seborrheic keratoses and piles	-	+	++
*Foeniculum vulgare* Mill., Apiaceae, UOP NK 00129	Kaga ^AI, AO, PO^ Badyan ^AI, AO^ Sonf ^PO^	Fruits ^AI, AO, PO^	FP ^AO, PO^ BM ^AI, AO, PO^	Mountains	M: powdered and then mixed in water, drunk as an anti-diarrheal and anti-ulcer, or sometimes powder is mixed with jaggery juice and drunk to treat bone pain and excessive perspiration, and to cool the body	++	++	+
*Fragaria vesca* L., Rosaceae, UOP NK 00130	Zangli strawberry ^AO^ zmakea tooth ^AO^	Fruits ^AO^	FP ^AO^	Fields	F: as raw snacks	-	+	-
*Fumaria indica* (Hausskn.) Pugsley, Papaveraceae, UOP NK 00131	Shatara ^AI, AO^ Papra ^AI^	Whole plants ^AI, AO^	FA ^AI, AO^ FP ^AI, AO^	Field edges	M: ground, filtrate is topically applied, or sometimes filtrate is mixed with water and drunk to treat skin rashes, cleanse blood, cool the body, and as an anti-itching agent	++	++	-
*Glycyrrhiza glabra* L., Fabaceae, UOP NK 00132	Khwagawolea ^AI, AO^	Roots ^AO^ Whole plants ^AI^	FA ^AI, AO^ BR ^AI, AO^ BM ^AO^	Mountains	B: decoction, anti-cough, and asthma V: fodder, heals internal body injuries *; roots powder and mixed with milk to heal bone fractures, and treat bone pain in goats, cows, and donkeys	+ *	+	-
*Gymnosporia royleana* Wall. ex M.A.Lawson, Celastraceae, UOP NK 00133	Surazghai ^AI^	Roots ^AI^	FP ^AI^	Mountains	M: decoction, anti-diabetic, and anti-cholesterol	+	-	-
*Gypsophila paniculata* L., Caryophyllaceae #	Badbarak ^AI, AO^	Whole plants ^AI, AO^	FA ^AI, AO^ BM ^AO^	Mountains	M: chewed to treat constipation and stomach gas; root powder is mixed with egg as an anti-pyretic and to heal wounds O: fruits are kept in wheat flour for prosperity (*barkat*); smoke as an anti-evil agent	+	++	-
*Juglans regia* L., Juglandaceae, UOP NK 00134	Ghoz ^AI, AO, PO^ Charmaghaaz ^AO^	Seeds ^AI, AO, PO^	FA ^AI, AO^ BR ^AI, AO^ BM ^AI, PO^	Mountains	F: seed, as raw snacks O: pericarp, use to brush teeth and color lips	+	++	-
*Justicia adhatoda* L., Acanthaceae, UOP NK 00135	Bekanrh ^AO^ Baza ^AO, PO^	Flowers ^AO^ Leaves ^AO, PO^	FP ^AO, PO^	Foothills	F: nectar is consumed fresh M: leaves are consumed fresh as a stomach coolant, or sometimes crushed leaves are mixed with water and drunk as an anti-diabetic, or fresh leaves are externally applied over pimples to extract pus O: leaves, fuel	-	++	+++
*Juniperus communis* L., Cupressaceae, UOP NK 00136	Obakhta ^AI, AO^	Fruit ^AI, AO^ Leaves ^AI^	FA ^AI, AO^ BR ^AI, AO^ BM ^AI^	Mountains	F: fruit, as raw snacks M: fruit, boiled in water as an anti-pyretic in children * V: leaves are placed on embers and the smoke used to remove intestinal worms; crushed fruits are mixed with wheat flour and ingested by animals to remove skin lesions *	+ *	++ *	-
*Lathyrus sativus* L., Fabaceae, UOP NK 00137	Zangli pali ^PO^	Seeds ^PO^	FP ^PO^	Fields and water banks	F: as raw snacks, or sometimes fried in oil	-	-	+
*Lepidium draba* L., Brassicaceae, UOP NK 00138	Bashka ^AI, AO, PO^	Leaves ^AI, AO, PO^	FA ^AI, AO^ FP ^AI, AO, PO^ BM ^AI^	Fields	F: boiled and then fried in oil with onions and other greens	++	++	++
*Lepidium sativum* L., Brassicaceae, UOP NK 00139	Sharghontai ^PO^ Halia ^PO^	Seeds ^PO^	FP ^PO^ BM ^PO^	Fields	M: decoction, anti-cough, and secondary amenorrhea	-	-	+
*Leuzea repens* (L.) D.J.N.Hind, Asteraceae #	Kuragh ^AI, AO^	Whole plants ^AI, AO^	FA ^AI, AO^	Fields and mountains	M: cold macerate (left for one night under the stars) drunk to treat gastrointestinal pain, hepatitis C, and diabetes, and to heal wounds, or sometimes crushed and stored in a big, closed container, in which the foot is placed for a short time, perceived as an anti-diabetic, and to treat heart pain and intestinal spasms V: fodder, or sometimes decoction, to remove intestinal worms and provide energy to camels *	++ *	++	-
*Linum usitatissimum* L., Linaceae, UOP NK 00140	Alsee ^PO^	Seeds ^PO^	FP ^PO^ BM ^PO^	Fields	M: powdered, half teaspoon of powder is mixed with water, perceived as anti-cough and anti-diabetic agents	-	-	+
*Malva neglecta* Wallr., Malvaceae, UOP NK 00141	Panderak ^AI, AO, PO^ Tikali ^AI, AO^ Puskai ^PO^	Leaves ^AI, AO, PO^ Roots ^AO^ Fruit ^AI^	FA ^AI, AO^ FP ^AI, AO, PO^ BM ^AO, PO^	Fields and graveyards	F: leaves, as salad, or sometimes fried in oil with other greens M: decoction, anti-arthritis * and to treat constipation V: decoction to reverse urine blockage *	++	++	+++
*Medicago arabica* (L.) Huds., Fabaceae, UOP NK 00142	Peshtaray ^AI, AO, PO^ Kunde sag ^AO, PO^ Akhwandak ^AI^	Aerial parts ^AI^ Leaves ^AO, PO^	FA ^AI, AO^ FP ^AI, AO, PO^ BM ^PO^ BR ^AO^	Wheat fields	F: aerial parts and leaves, boiled and then fried in oil, or sometimes ground, then later boiled, and cooked with garlic, ginger, chilies, and tomatoes M: effective against intestinal problems	++	++	+++
*Melia azedarach* L., Meliaceae, UOP NK 00143	Bakarra ^AI, AO^	Roots ^AI^ Fruits ^AI, AO^	FP ^AO^ FA ^AI, AO^	Fields	O: root, filtrate is mixed with a decoction of walnut bark, added to henna, and then externally applied on body hair as a coloring agent *; fruit, powdered and mixed with milk or water and drunk to boost energy *, or sometimes fruit, powder is mixed with oil and externally applied over hair as an anti-lice agent	+ *	+	-
*Mentha longifolia* (L.) L., Lamiaceae, UOP NK 00144	Shinshubaye ^AI, AO^ Venalle ^AI^ Zangli podina ^PO^ Villanai ^AO^	Leaves ^AO, PO^ Aerial parts ^AI^	FA ^AI, AO^ FP ^AI, AO, PO^ BM ^AI^	Stream banks	F: leaves, as raw snacks, or sometimes used to make sauce (*chatni*) M: cold macerate (let macerate for one night under sky/stars and drunk before breakfast) as an anti-diabetic, to treat digestive problems, high blood pressure, stomachache, intestinal pain, and as an anti-diarrheal in children O: dried leaves, as a spice	++	+++	+++
*Monotheca buxifolia* (Falc.) A.DC., Sapotaceae, UOP NK 00145	Gurgura ^AI, AO, PO^	Fruits ^AI, AO, PO^ Leaves ^AO^	FA ^AI, AO^ FP ^AI, PO^ BM ^AI, AO, PO^	Mountains	F: fruits, as raw snacks V: leaves, as fodder	+	++	+++
*Morchella esculenta* Fr., Morchellaceae, UOP NK 00146	Tur sture ^AI^ Sheesha ^AI^ Starii ^AO^ Gargichoo ^AO^ Gochi ^AO^ Karkichoo ^PO^	Aerial parts ^AI, AO, PO^	FA ^AI, AO^ BR ^AI, AO^ BM ^AI, AO, PO^	Stream banks and mountains	F: fried in oil M: filtrate, few drops in the eyes improve eyesight *, or sometimes dried powder is mixed with water and drunk to treat cholera	++ *	++	+
*Morus alba* L., Moraceae, UOP NK 00147	Khasak toot ^AO^ Spin toot ^AI, AO, PO^	Fruits ^AI, AO, PO^ Leaves ^PO^	FA ^AI, AO^ FP ^AI, AO, PO^	Fields	F: fruits, as raw snacks M: infusion, anti-cough, antipyretic, and to treat flu V: leaves, as fodder, anti-diarrheal and anti-pyretic *	++ *	++	+++
*Morus nigra* L., Moraceae, UOP NK 00148	Tur toot ^AI, AO, PO^	Fruits ^AI, AO, PO^ Leaves ^PO^	FA ^AI, AO^ FP ^AI, AO, PO^	Fields	F: fruit, as raw snacks M: infusion, anti-cough, antipyretic, and to treat flu V: leaves, as fodder, anti-diarrheal and anti-pyretic *	++ *	++	+++
*Myrtus communis* L., Myrtaceae, UOP NK 00149	Manrho ^AI, AO^ Manrhogan ^AI^	Fruits ^AI, AO^ Leaves ^AI, AO^	FA ^AI, AO^ BR ^AO^	Mountains	F: fruit, as raw snacks, or sometimes used to make tea M: infusion, as an appetite booster O: leaves, burnt on embers, the smoke is considered anti-evil eye and anti-ghost agents, and as an air freshener in homes	++	++	-
*Nannorrhops ritchieana* (Griff.) Aitch., Arecaceae, UOP NK 00150	Mazare ^AI^	Fruits ^AI^ Leaves ^AI^	FA ^AI^ FP ^AI^	Mountains	F: fruits, as a raw snack V: infusion, to treat intestinal pain in cows and buffalos	++	-	-
*Nasturtium officinale* W.T.Aiton, Brassicaceae, UOP NK 00151	Tarmeera ^AI, AO, PO^ Jamia ^AO^ Zanglijamia ^PO^ Jarjir ^PO^	Leaves ^AI, AO, PO^ Seeds ^PO^	FA ^AI, AO^ FP ^AI, AO, PO^ BM ^AO, PO^	Fields and stream banks	F: leaves, fried in oil with onions and other greens, or sometimes fresh leaves are consumed as a salad M: seeds, the extracted oil is orally taken to treat stomach gas and joint pain V: seeds, as fodder and an anti-parasitic	+	++	+++
*Nepeta laevigata* (D.Don) Hand.-Mazz., Lamiaceae, UOP NK 00152	Ogratengea ^AI^ Gulbakhor ^AI, AO^	Whole plants ^AI^ Aerial shoots ^AO^	FA ^AI, AO^	Fields, foothills, and mountains	M: decoction, drunk before breakfast and after dinner to treat stomach gas, muscle contractions, cardiac pain, malaria, typhoid, fever, cough, constipation, and as an anti-diarrheal in children, or sometimes women drink to treat gastric troubles and to increase ovulation *; cold macerate, anti-pyretic and anti-cough V: decoction, decrease abdominal distension caused by constipation *	+++ *	++	-
*Nerium oleander* L., Apocynaceae, UOP NK 00152	Gandrai ^AO^ Gandezarai ^PO^ Kanair ^PO^	Leaves ^AO, PO^	FP ^AO, PO^	Fields and mountains	M: paste, applied over pimples in children, or sometimes used to brush teeth to treat toothache; sometimes leaves together with *Fegonia indica* are crushed, extract is applied over pimples and itchy skin	-	+	++
*Nymphaea* sp., Nymphaeaceae #	Barsanda ^AO^	Aerial parts ^AO^	FA ^AO^ BM ^AO^	Fields	F: fried in oil with onions, tomatoes, and chilies	-	+	-
*Ocimum basilicum* L., Lamiaceae, UOP NK 00153	Tukhmeamalang ^PO^ Kashmali ^PO^ Naizbow ^PO^	Leaves ^PO^	FP ^PO^	Fields	M: chewed, for mouth dryness and good fragrance	-	-	++
*Olea europaea* subsp. cuspidata (Wall. and G.Don) Cif., Oleaceae, UOP NK 00154	Khowan ^AI, AO^ Zytoon ^AI^ Kacha zytoon ^AO^ Shwan ^PO^	Leaves ^AI, AO, PO^ Fruits ^PO^ Stems ^AO, PO^	FA ^AI, AO^ FP ^AI, AO, PO^	Mountains	F: leaves, to make recreational tea M: leaves, powder is orally taken with water to treat diabetes; fruits, extracted oil is topically applied to massage the body V: leaves, as fodder O: fruits, extracted oil is used as cooking oil; stem wood is used to make ax hafts	+	++	+++
*Opuntia triacanthos* (Willd.) Sweet, Cactaceae, UOP NK 00155	Zaqqoom ^AI^	Stems ^AI^ Fruits ^AI^	FA ^AI^ FP ^AI^	Fields	M: stem extract, externally applied on hair to treat hair loss and skin problems *; fruit, eaten to treat diabetes and high blood pressure	++ *	-	-
*Oxalis corniculata* L., Oxalidaceae, UOP NK 00156	Troshkai ^AI^ Tarveeka ^AO^	Leaves ^AI, AO^	FA ^AI,^ FP ^AI, AO^	Fields, shady places, and water banks	F: fried in oil with other greens, or sometimes also used to make sauces (*chatnii*)	+	+	-
*Papaver rhoeas* L., Papaveraceae.#	Reday gul ^AI^	Whole plants ^AI^	FA ^AI^	Mountain, and different crops	F: boiled and then fried in oil with other greens	+	-	-
*Parthenium hysterophorus* L., Asteraceae, UOP NK 00157	Ghanda bote ^AI^	Whole plants ^AI^	FP ^AI^	Fields and mountains	M: powdered, then mixed with water as an anti-diabetic	+	-	-
*Peganum harmala* L., Nitrariaceae, UOP NK 00158	Spilanai ^AI, AO, PO^ Sponda ^AO^	Leaves ^AI^ Seeds ^AO, PO^	FA ^AI, AO, PO^ FP ^AI, AO, PO^ BM ^AI, AO^	Mountains	M: leaves, powdered and mixed with oil to massage body, perceived as an anti-diabetic; decoction of *Peganum harmala*, *Trigonella foenum*, and *Lepidium sativum* as menses inducers in women O: seeds, sprinkled on embers, the smoke considered an anti-evil eye agent, and snake and scorpion repellent	++	+++	+++
*Periploca aphylla* Decne., Apocynaceae, UOP NK 00159	Barara ^AI, AO^	Gum ^AO^ Stem bark ^AI^	FA ^AI, AO^ FP ^AI^	Mountains	F: gum, chewed M: decoction, mixed with mother milk as an appetite suppressant in children	+	+	-
*Pinus gerardiana* Wall. ex D.Don, Pinaceae, #	Zanrghozai ^AI, AO^	Seeds ^AI, AO^ Resin ^AI, AO^	FA ^AI, AO^	Mountains	F: seeds, as raw snacks M: resin, externally applied to heal wounds	+	+	-
*Pinus roxburghii* Sarg., Pinaceae, #	Nakhtar ^AO, PO^	Resin ^AO^ Gum ^AO, PO^	BM ^PO^ BR ^AO^ FA ^AO^	Mountains	M: resin, 2 drops are mixed in yogurt to remove pimples and to treat hyperthermia; gum, chewed to boost energy by pregnant females; also chewed to clean the teeth and stop itching	-	+	+
*Pistacia khinjuk* Stocks, Anacardiaceae, #	Shinyea ^AI^ Pista ^AO^	Fruits ^AI^ Seeds ^AO^	FA ^AO^ BM ^AO^	Mountains	F: fruits and seeds, as raw snacks	+	+	-
*Plantago lanceolata* L., Plantaginaceae, UOP NK 00160	Ghoizaba ^AI, AO, PO^	Leaves ^AI, AO, PO^	FA ^AI, AO^ FP ^AI, AO, PO^	Fields	F: fried in oil with other greens M: filtrate, few drops in eyes to treat conjunctivitis; infusion to treat chest pain in children V: leaves, as fodder	++	++	+
*Plantago major* L., Plantaginaceae, UOP NK 00161	Bartaang ^AI, AO, PO^	Seeds ^AI, AO, PO^	FA ^AI, AO^ FP ^AI, PO^ BM ^AO, PO^	Fields, wetlands, and stream sides	M: seeds, used in different syrups to treat constipation and stimulate appetite in children; decoction, antipyretic, anti-cough, and to treat chest pain in children; seeds, mixed with milk to treat intestinal pain in children	++	++	++
*Plantago ovata* Forssk., Plantaginaceae, UOP NK 00162	Sat ^AI^ Ispeghol ^AI, AO, PO^	Seeds ^AO^ Whole plants ^AI^	FA ^AO^ FP ^AO^ BM ^AO^	Fields	M: seeds, as a constituent cooling agent in syrups, or sometimes mixed with yogurt and banana, then ingested to treat constipation and to cool the stomach; whole plant is crushed, powder is mixed with boiled milk to treat stomach problems and to decrease blood cholesterol level	++	+	+
*Polygonum verticillatum* Biroli ex Colla, Asparagaceae, UOP NK 00163	Miralam ^AO^ Nooralam ^PO^	Roots ^AO, PO^	FA ^AO^ BM ^PO^ BR ^AO^	Mountains	M: fried in oil and consumed to treat chest wounds, perceived as an aphrodisiac, and to heal bullet wounds *	-	++ *	++
*Portulaca oleracea* L., Portulacaceae, UOP NK 00164	Warkharea ^AI, AO, PO^ Kulfa ^PO^	Aerial parts ^AI, AO, PO^	FA ^AI, AO^ FP ^AI, AO, PO^ BM ^AI, AO, PO^	Fields	F: fried in oil with onions, tomatoes, and chilies	++	++	+++
*Prosopis juliflora* (Sw.) DC., Fabaceae, UOP NK 00165	Lewanai kekar ^AO, PO^	Gum ^AO^ Leaves ^AO^ stems ^PO^	FP ^AO, PO^	Mountains and fields	F: gum, eaten M: leaves, paste topically applied to reduce pain and inflammation caused by thorns O: stems, as fuel wood	-	+	++
*Prunus amygdalus* Batsch, Rosaceae #	Zangli badam ^AI^	Seeds ^AI^	FA ^AI^	Garden, mountains, and streamside	F: as raw snacks M: powdered, 2–3 teaspoons of powder are mixed with milk to treat brain weakness and dermatitis	+	-	-
*Prunus armeniaca* L., Rosaceae #	Zangli mandata ^AI, AO^	Fruits ^AI, AO^ Gum ^AI, AO^	FA ^AI, AO^	Mountains	F: fruits, as raw snacks M: gum (*kund*), boiled in milk or water and drunk before breakfast to treat back aches	++	++	-
*Prunus jacquemontii* Hook.f., Rosaceae #	Arghanjea ^AO^	Fruit ^AO^	FA ^AO^ BR ^AO^	Fields	F: fruits are boiled, filtered, and crushed to make jam	-	+	-
*Punica granatum* L., Lythraceae, UOP NK 00166	Annar ^AI, AO, PO^	Fruits ^AI, AO, PO^ Roots ^AI^ Seeds ^AI^	FA ^AI, AO^ FP ^AI^ BM ^PO^	Mountains	F: fruits, as raw snacks M: decoction (root), to treat ulcers and to heal internal wounds caused by sugar and excess use of red chilies *; exocarp and seeds are chewed to improve digestion *; exocarp is heated on a metal plate (*Teghna*) and then crushed to make a powder which is taken with water to treat constipation and diarrhea V: exocarp as fodder, or sometimes powdered and mixed with water and then given to animals to treat constipation and diarrhea	++ *	++	+
*Quercus incana* W.Bartram, Fagaceae, UOP NK 00167	Pargai ^AI, AO^ Serea ^AI, AO, PO^	Fruits ^AI, AO, PO^ Stem ^AO, PO^	BM ^PO^ BR ^AI, AO^ FA ^AI, AO^	Mountains	F: fruits, fried on a metal plate (*teghna*) and ingested, or sometimes mixed with maize bread and eaten M: effective to treat cardiovascular complications and as an anti-cholesterol V: fruits, as fodder in winter O: stems, as fuel wood	++	++	+
*Rheum australe* D.Don, Polygonaceae #	Keraskai ^AI, AO^	Roots ^AI, AO^	FA ^AI, AO^ BM ^AO^	Mountains	M: decoction, to heal wounds and to relieve intestinal pain *, or sometimes root powder is mixed with milk and drunk to treat internal wounds, bone fractures, and back aches	+ *	+	-
*Rheum rhabarbarum* L., Polygonaceae #	Sukrai ^AI, AO^ Pakhea ^AI^	Stem ^AO^ Gel ^AI^	BM ^AI^ FA ^AI, AO^	Mountains	F: stems, as raw snacks M: gel, consumed directly as a source of energy *	+ *	+	-
*Rheum ribes* L., Polygonaceae #	Chukri ^AI, AO^ Rawash ^AO^ Pshai ^AO^	Leaves ^AI, AO^	FA ^AI, AO^	Fields and mountains	F: boiled, then fried in oil with onions, tomatoes, chilies, and other greens, or sometimes dried leaves along with wheat and barley flour are mixed with whig and then eaten	+	++	-
*Ricinus communis* L., Euphorbiaceae, UOP NK 00168	Arand ^PO^	Seeds ^PO^ Leaves ^PO^	FP ^PO^ BM ^PO^	Fields and water banks	M: seeds, directly taken before menses to avoid pregnancy, and against paralysis; leaves are kept on henna which causes cooling and color ripening O: leaves, kept on henna which is used for color ripening	-	-	++
*Rosa* sp., Rosaceae *#*	Zangli Gulab ^AI, PO^	Flower ^AI, PO^	FP ^AI, PO^	Home gardens and mountains	M: powder, mixed with water and drunk to treat stomach aches and as an anti-diabetic, or sometimes flowers are heated on a metal plate and kept for 8–10 days to prepare gulkand and drunk to treat constipation in pregnant women O: gulkand, acts as a galactagogue	++	-	+
*Rubus fruticosus* L., Rosaceae #	Nanga ^AO^	Fruits ^AO^	FA ^AO^	Mountains	F: fruit, as raw snacks	-	+	-
*Rumex dentatus* L., Polygonaceae, UOP NK 00169	Shalkhay ^AI, AO, PO^ Zanda ^AI, AO, PO^ Papar ^PO^	Leaves ^AI, AO, PO^	FA, ^AI, AO^ FP ^AI, AO, PO^	Fields and water banks	F: boiled and then fried in oil with onions, tomatoes, and other greens	++	++	+++
*Sageretia thea* (Osbeck) M.C.Johnst., Rhamnaceae #	Mamanra ^AI, AO, PO^	Fruits ^AI, AO^ Roots ^AI, PO^	FA ^AI, AO^ FP ^AI, AO, PO^ BM ^PO^	Mountains	F: as raw snacks M: decoction (roots) to treat hepatitis B and C, and typhoid	++	++	++
*Salvadora oleoides* Decne., Salvadoraceae, UOP NK 00170	Plawan ^AI, PO^ Pleen ^PO^	Fruits ^AI^ Whole plants ^PO^	FA ^AI,^ FP ^AI, PO^	Graveyards and fields	F: fruits, as raw snacks O: whole plant, as fuel wood	+	-	++
*Salvadora persica* L., Salvadoraceae, UOP NK 00171	Miswak ^AI, AO^	Roots ^AI, AO^	FA ^AI, AO^ FP ^AO^	Graveyards and fields	O: as a toothbrush (*Maswak*) to clean teeth	+	++	-
*Salvia nubicola* Wall. ex Sweet, Lamiaceae #	Darshool ^AO^	Seeds ^AO^	FA ^AO^	Fields	M: pod is cut, and one piece is taken orally to treat dizziness	-	+	-
*Salvia yangii* B.T.Drew, Lamiaceae, UOP NK 00172	Sansubai ^AI, AO^	Leaves ^AI^ Aerial parts ^AO^	FA ^AI, AO^ BM ^AI^	Fields and mountains	M: leaves, placed over embers and the smoke is perceived as remedy for fever and intestinal pain *, or sometimes leaves are topically applied over wounds to stop bleeding O: flowers consumed by honeybees to produce better quality honey; leaves, as fuel; and aerial parts are dropped in the well to make the water pleasant and clean *	++ *	++ *	-
*Scorpiurus muricatus* L., Fabaceae, #	Spin saba ^AO^	Leaves ^AO^	FA ^AO^	Foothills	F: boiled and then fried in oil with onions, chilies, and tomatoes	-	+	-
*Senegalia modesta* (Wall.) P.J.H.Hurter, Fabaceae, UOP NK 00173	Palosa ^AO, AI, PO^	Gum ^AO, AI, PO^, Stem bark ^PO^	FA ^AO^ FP ^AO, AI, PO^ BM ^AI, PO^	Fields and mountains	F: gum, directly taken with dry fruits or mixed with other confections M: one teaspoon of gum is dissolved in either a glass of milk, water, or sometimes oil to treat joint and back pain; stem bark (oil extract) is topically applied to treat joint pain V: one teaspoon of gum is dissolved in either a glass of milk, water, or sometimes oil to treat intestinal wounds in animals * O: stems, as fuel wood, or sometimes wood is used to make weapon parts *; oil is also used for fragrance	++ *	++	+++
*Solanum nigrum* L., Solanaceae, UOP NK 00174	Kachmachoo ^AI, AO^	Fruits ^AI, AO^ Leaves ^AO^	FA ^AI, AO^ FP ^AI, AO^	Fields and mountains	F: fruits, as raw snacks; leaves, fried in oil with other greens O: fruits, topically applied by women to make *sheen khaal* (blue-green permanent dotted marks and patterns usually tattooed between the eyebrow)	+	++	-
*Solanum villosum* Mill., Solanaceae #	Makan botai ^AI^	Leaves ^AI^ Stems ^AI^	FP ^AI^	Mountains	M: leaves and stems, powdered and mixed with milk and drunk as an aphrodisiac	+	-	-
*Spinacia oleracea* L., Amaranthaceae, UOP NK 00175	Palak ^AI, AO^	Leaves ^AI, AO^	FA ^AI, AO^ FP ^AI, AO^	Fields	F: boiled and then fried in oil with tomatoes and onions	++	++	-
*Tamarix aphylla* (L.) H.Karst., Tamaricaceae, UOP NK 00176	Ghaz ^AI, AO, PO^	Stem bark ^AI, AO^ Leaves ^PO^	FA ^AI^ FP ^AI, AO, PO^	Fields and sandy lands	M: stem bark, powder, directly or mixed with fried oil, is topically applied to heal burns and wounds; decoction, gargle 2–3 times to treat toothache; leaves, boiled, one drop of filtrate is mixed with water to treat diabetes; leaves, chewed to heal internal wounds	+	+	+++
*Teucrium stocksianum* Boiss., Lamiaceae, UOP NK 00177	Kasterai ^AI^ Mastyera ^PO^	Leaves ^AI^ Flowers ^AI^ Whole plants ^PO^	FA ^AI^ FP ^AI, PO^	Mountains	M: leaves and flowers, powdered and mixed with water to treat cardiac problems; filtrate, anti-diabetic, antipyretic, and for cooling the body	+	-	+++
*Thymus* sp., Lamiaceae #	Panai ^PO^ Mawruza ^AI, AO^	Whole plants ^AI, AO, PO^	BM ^PO^ FA ^AI, AO^	Mountains	F: leaves, consumed as a salad; whole plant, powdered and used to make recreational tea M: flowers, powdered and orally taken to treat constipation and bloating O: flowers, powder is kept in the mouth to cure snuff addiction	+	++	+
*Tinospora cordifolia* (Willd.) Hook.f. and Thomson, Menispermaceae, UOP NK 00178	Gillow ^PO^	Young shoots ^PO^	FP ^PO^ BM ^PO^	Fields and foothills	M: decoction, blood purification	-	-	+
*Trachyspermum ammi* (L.) Sprague, Apiaceae, UOP NK 00179	Sperkai ^AI, AO, PO^ Jowani ^AO^ Ajwan ^PO^	Fruits ^AI, AO, PO^	FA ^AI, AO^ FP ^AI, PO^ BM ^PO^	Fields, crop fields, and mountains	M: fried on a metal plate (*Teghna*), mixed with black tea and poppy plant, and then drunk to treat cough and phlegm, or sometimes powder is mixed with water and drunk to treat diarrhea, or sometimes seeds are added to jaggery syrup and put in a piece of cloth and dipped in milk which is drunk to treat high fever caused by anhidrosis	++	+++	+
*Trigonella foenum-graecum* L., Fabaceae, UOP NK 00180	Methrai ^AO^ Shambrhetea ^PO^ Methidana ^PO^	Seeds ^AO, PO^ Leaves ^PO^	FA ^AO^ FP ^AO, PO^ BM ^PO^	Fields and mountains	M: seeds, one teaspoon is mixed with water to treat diabetes and body pain O: leaves, as a spice	-	+	++
*Tulipa* sp., Liliaceae, #	Shamdai ^AI^	Bulbs ^AI^	FA ^AI^ FP ^AI^	Mountains	F: consumed as raw snacks	+	-	-
*Urtica dioica* L., Urticaceae, UOP NK 00181	Sizunkea ^AI^	Whole plant ^AI^	FP ^AI^	Mountains	F: fried in oil with onions and tomatoes	+	-	-
*Vachellia nilotica* (L.) P.J.H.Hurter and Mabb., Fabaceae, UOP NK 00182	Kikar ^PO^	Gums ^PO^ Fruits ^PO^	FP ^PO^	Fields	F: gum, eaten directly V: fruits, as fodder	-	-	++
*Valeriana jatamansi* Jones ex Roxb., Caprifoliaceae, #	Makhkak ^AO^	Rhizomes ^AO^	FA ^AO^	Mountains	O: burned and the smoke is considered a myth to reduce poverty in homes	-	+	-
*Verbascum* sp., Scrophulariaceae #	Zakhta ^AI, AO^ Papoka ^AI^	Roots ^AI^ Leaves ^AO^	FA ^AI^	Fields and mountains	F: roots, as raw snacks * M: leaves, powdered and mixed with water to treat intestinal and chest problems	+ *	+	-
*Verbascum thapsus* L., Scrophulariaceae, UOP NK 00183	Khar kharghwag ^AI, AO^	Whole plants ^AI^ Leaves ^AO^	FA ^AI, AO^	Fields and mountains	V: filtrate, increases fat and meat, which later also minimizes pain during delivery * O: leaves, boiled in water and later used as detergent *	++ *	++ *	-
*Vitis vinifera* L., Vitaceae, UOP NK 00184	Zangli angoor ^AI^ Angoor ^AO^ Manaka ^AO^	Fruits ^AI, AO^	FA ^AI, AO^ BM ^AO^	Mountains	F: as raw snacks	+	++	-
*Withania coagulans* (Stocks) Dunal, Solanaceae, UOP NK 00185	Khamzoora ^AI, AO^ Shapyanga ^AO, PO^ Mahori ^PO^ Spinabaza ^PO^	Leaves ^AI, AO^ Fruits ^AI, AO, PO^	FA ^AI, AO^ FP ^AI, AO, PO^ BM ^AI, AO, PO^	Fields and mountains	M: infusion, to treat hepatitis, also helps with digestive disorders; fruits, taken directly with water to treat diarrhea, stomach gas, gastric pain, diabetes, coolness, body pimples, and cardiac pain V: infusion is mixed with some jaggery to treat colds O: fruits, powder is put in a piece of cloth and dipped in milk for fermentation	+++	+++	+++
*Withania somnifera* (L.) Dunal, Solanaceae, UOP NK 00186	Azgand ^PO^	Roots ^PO^ Stems ^PO^	FP ^PO^ BM ^PO^	Fields and mountains	M: roots and stems, powdered and orally taken with water to strengthen to bone marrow	-	-	+
*Ziziphus muratiana* Maire, Rhamnaceae, UOP NK 00187	Sawa bera ^PO^	Fruits ^PO^	FP ^PO^	Fields	F: fruits, as raw snacks	-	-	++
*Ziziphus jujuba* Mill., Rhamnaceae, UOP NK 00188	Ghata Berrha ^AI^ Berrha ^AO, PO^	Fruit ^AI, AO, PO^ Roots ^AI^ Leaves ^PO^ Flowers ^AO^	FA ^AI^ FP ^AI, AO, PO^	Fields and mountains	F: fruits, as raw snacks M: fruits, to stop pre-ejaculatory fluid; decoction, as anti-cough V: leaves, as fodder and anti-diarrheal O: flowers act as a source of nectar and pollen for honeybees to produce better quality honey	+	++	+++
*Ziziphus nummularia* (Burm.f.) Wight and Arn., Rhamnaceae, UOP NK 00189	Kankara ^AI, AO, PO^	Fruits ^AI, AO, PO^ Roots ^AI^ Root bark ^AO^	FA ^AI^ FP ^AI, AO, PO^	Fields and mountains	F: fruits, as raw snacks M: decoction, to treat cough and hepatitis C V: fruits, as fodder	+	++	+++
*Zygophyllum indicum* (Burm.f.) Christenh. and Byng, Zygophyllaceae, UOP NK 00190	Azghakai ^AI, AO^ Shinazghai ^AO^ Mazhgakai ^PO^	Whole plants except roots ^AI, AO, PO^	FA ^AI, AO^ FP ^AI, AO, PO^ BM ^AI^	Fields, foothills, and mountains	M: cold macerate, to treat malaria, typhoid fever, hyperthermia, kidney pain, itching, bone aches, as anti-cough anti-diabetic agents, for blood purification, and as an aphrodisiac; extract is topically applied as an anti-pimple agent	+++	+++	+++

#: Identification carried out via pictures, folk names, and plant descriptions only; F: food use; V: ethno-veterinary use; M: medicinal use; O: other use; FA: foraged in Afghanistan; FP: foraged in Pakistan; BM: ingredients bought at the local markets in Pakistan; BR: brought from the Afghanistan; AI: Afghan refugees living inside the camps, AO: Afghan refugees living outside the camps, PO: local Pashtuns living outside the camps, *: use remembered from the past (back 40 years); + rarely quoted (less than or equal to 10% of study participants); ++ commonly quoted (11–40% of study participants); +++ very commonly quoted (more than 40% of study participants).

**Table 2 plants-12-00574-t002:** Taxa quoted by the three considered groups.

GROUP	AI	AO	PO
Wild food plants (number and % of imported species from Afghanistan)	46 taxa (6/4.5%)	61 taxa (13/9.8%)	39 taxa
Medicinal plants (number and % of imported species from Afghanistan)	67 taxa (11/8.3%)	58 taxa (16/12%)	37 taxa
Veterinary plants (number and % of imported species from Afghanistan)	16 taxa (4/3.0%)	10 taxa (5/3.8%)	12 taxa
Total (number and % of imported species from Afghanistan)	96 taxa of which (12/9.0%)	98 taxa of which (20/15%)	70 taxa

**Table 3 plants-12-00574-t003:** Characteristics of the study participants.

Groups	Afghan Pathans Inside the Refugee Camps (AI)	Afghan Pathans Outside the Refugee Camps (AO)	Local Pathans (PO)
Arrival of the studied community in the area	1979–1981	Possibly 13th–15th centuries A.D.	
Number of study participants	28 male/2 female	27 male/3 female	24 male/6 female
Average age	63 years	60 years	65 years
Religious faith	Sunni Muslim		
Ecology of the area		Foothills	Plain and mountains
Traditional intermarriage rules	Only with other Afghans	Only with other local Pathans	
Main traditional and modern occupation	Pastoralism, shopkeeping, employees	Pastoralism, horticulturalism	Horticulturalism, employees (also in governmental offices), shopkeeping
Language	Pashto (Zazai, Shinwari, Karlani, Zadrani, Apridi, Bungish, and Khattak dialects)		
Main tribes to which the study participants belong	Shinwari, Zadran, Khogyani, and Zazi	Afridi, Bangash, and Khattak	
Estimated socio-economic status	Low	Middle	Middle/high

## Data Availability

All data supporting the results of this research are included within the article.

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
