# Peer review of "Disadvantaged Economic Conditions and Stricter Border Rules Shape Afghan Refugees’ Ethnobotany: Insights from Kohat District, NW Pakistan"

_plants, 2023, doi:10.3390/plants12030574_

Round 1

Reviewer 1 Report

This manuscript deals with plant traditional use by three groups of Afghan and Pakistan people sharing the language, but living in three clearly different situations. It is a very relevant contribution to migrants’ (compared with non-migrant’s) ethnobotany. The work is correctly designed, the methods seem to have been adequately applied and the results are numerous and of high quality, and are discussed in the frame of a pertinent literature set, which is logical, according to the long experience in this subject (ethnobotany and particularly migrants’ one) of some of the co-authors. I believe the manuscript is clearly worthy of publication, and I suggest it to be accepted after a minor revision, for which I provide now the authors with some comments.

1.- The first observation is just a reflection related to the statement of the authors themselves on the scarce number of female informants (I noted “why so few women?!” in the document I was reading, before reaching this statement). I am aware that the situation described by the authors (in the human groups prospected, it is not well seen that a foreign man talks with a local female) is as described. Of course, I am sure that, if, among the authors, one woman had been able to talk the local language, this bias would have been smaller, but I understand that this has not been possible. In any case, this situation does not at all diminish the interest of the current manuscript, while it opens the door to a completion with more women’s data whenever possible.

2.- Please do not writhe with capital initials terms such as local ecological knowledge or wild food plants (and some others), even if they have an acronym, in the same way we write deoxyribonucleic acid and not DeoxyRibonucleic Acid, even if the acronym DNA is used.

3.- Delete the dot after m as abbreviation of meter, since it is an international measurement symbol.

4.- In the territory description, a short explanation of the plant landscape would be useful.

5.- Since the field work was performed in 2022, a comment on the changes due to relatively recent return of an ancient regime in Afghanistan, if any, should be commented when changes regarding refugees are stated.

6.- The authors writhe “Ninety participants were conveniently selected”. What ‘conveniently’ means in this case should be detailed.

7.- The authors state that ‘raw snacks’ and ‘cooked vegetables’ are the most reported kinds of food plant uses. Later, they mention ‘salads’. It is not clear to me whether raw snacks include salads (intuitively I would say no, but I am not convinced of the authors’ use reading the manuscript).

8.- Dying hairs in black is considered as a medicinal use, but I believe it would be better treated as a cosmetic use.

9.- The authors write “According to the study participants, most often ethnoveterinary knowledge is learned from elderly individuals and is passed to the next generation from father to child, but, in certain cases, it may be learned from friends or neighbors through general group discussions”. If I am not wrong, no similar comments are provided for other kind of uses than ethnoveterinary, and I believe that it would also be pertinent to reflect on this regarding other uses.

10.- Similarly the authors write “Nearly one-fourth of the ethnoveterinary uses were reported to be practiced in the past…”, but I do not see similar comments (which I would find interesting) for other kinds of uses.

11.- When the texts are in italics (textual quotations), please do not italicise scientific names.

12.- An important part of the results’ section is for me rather discussion, as numerous comparisons and commonalities’ analyses.

13.- The authors write “The above statement suggests that the progressive weakening of the plant-people relationship between the older and younger generations could probably further lead to the erosion of LEK”, referring to current plant use. Did they evaluate, apart from the use, the remembrance of informants of uses that were common in preceding generations? In case this corpus of knowledge was still relevant, it could make the erosion lower.

14.- Replace ‘boarder’ with ‘border’.

15.- The authors state that in some cases the uses are quoted by a small number of people. Related to this, and in general, I believe that calculating some indices (such as cultural value of informant’s consensus factor) would strengthen the analysis and the discussion of the results.

16.- In table 1, please specify the name of the local language in column 3’s heading.

17.- In table 1, three solutions appear for taxa undetermined at specific level, such as Allium ssp., Thymus sp., Nymphaea L. In the first case, I could understand several Allium species (but it is not clear, the local name being common). In the second case, I understand an undetermined Thymus species. In the third case, for Nymphaea, does this mean all species of the genus?

18.- In table 1, Artemisia ssp. should probably be replaced with Artemisia scoparia, and Tulip sp. should be replaced with Tulipa sp.

Author Response

Dear Reviewer,

Thank you for your meaningful feedback on our manuscript. We replied to each of the points you raised below. 

This manuscript deals with plant traditional use by three groups of Afghan and Pakistan people sharing the language, but living in three clearly different situations. It is a very relevant contribution to migrants’ (compared with non-migrant’s) ethnobotany. The work is correctly designed, the methods seem to have been adequately applied and the results are numerous and of high quality, and are discussed in the frame of a pertinent literature set, which is logical, according to the long experience in this subject (ethnobotany and particularly migrants’ one) of some of the co-authors. I believe the manuscript is clearly worthy of publication, and I suggest it to be accepted after a minor revision, for which I provide now the authors with some comments.

The first observation is just a reflection related to the statement of the authors themselves on the scarce number of female informants (i noted “why so few women?!” in the document i was reading, before reaching this statement). I am aware that the situation described by the authors (in the human groups prospected, it is not well seen that a foreign man talks with a local female) is as described. of course, i am sure that, if, among the authors, one woman had been able to talk the local language, this bias would have been smaller, but i understand that this has not been possible. In any case, this situation does not at all diminish the interest of the current manuscript, while it opens the door to a completion with more women’s data whenever possible.

We have better articulated the issue of gender in the Methods.

2.- Please do not write with capital initials terms such as local ecological knowledge or wild food plants (and some others), even if they have an acronym, in the same way we write deoxyribonucleic acid and not DeoxyRibonucleic Acid, even if the acronym DNA is used.

Thank you for noticing, we have corrected the acronyms throughout the text.

3.- Delete the dot after m as an abbreviation of metre, since it is an international measurement symbol.

Thank you for noticing, we have corrected this throughout the text.

4.- In the territory description, a short explanation of the plant landscape would be useful.

We have added a paragraph on this in the Materials section.

5.- Since the field work was performed in 2022, changes due to the relatively recent return of an ancient regime in Afghanistan, if any, should be commented on when changes regarding refugees are stated.

A paragraph has been added on this in the Discussion.

6.- The authors write “Ninety participants were conveniently selected”. What ‘conveniently’ means in this case should be detailed.

We have now clarified this issue in the Methods.

7.- The authors state that ‘raw snacks’ and ‘cooked vegetables’ are the most reported kinds of food plants use. Later, they mention ‘salads’. It is not clear to me whether raw snacks include salads (intuitively I would say no, but I am not convinced of the authors’ use reading the manuscript).

We have clarified this in the Methods section.

8.- Dying hairs in black is considered as a medicinal use, but I believe it would be better treated as a cosmetic use.

We have better placed it in the category “others” which includes cosmetic uses.

9.- The authors write “According to the study participants, most often ethnoveterinary knowledge is learned from elderly individuals and is passed to the next generation from father to child, but, in certain cases, it may be learned from friends or neighbours through general group discussions”. If I am not wrong, no similar comments are provided for other kinds of uses than ethnoveterinary, and I believe that it would also be pertinent to reflect on this regarding other uses.

We have specified these concepts in both the medicinal and food sections.

10.- Similarly the authors write “Nearly one-fourth of the ethnoveterinary uses were reported to be practiced in the past…”, but I do not see similar comments (which I would find interesting) for other kinds of uses.

We have specified these concepts in both the medicinal and food sections.

11.- When the texts are in italics (textual quotations), please do not italics scientific names.

We have corrected this.

12.- An important part of the results’ section is for me rather than discussion, as numerous comparisons and commonalities’ analyses.

Actually, these points answer the second objective “to identify commonalities and differences in the folk plant uses among the three groups and to possibly interpret these findings in cultural terms”. Therefore, we would like to keep them as part of the results.

13.- The authors write “The above statement suggests that the progressive weakening of the plant-people relationship between the older and younger generations could probably further lead to the erosion of LEK”, referring to current plant use. Did they evaluate, apart from the use, the remembrance of informants of uses that were common in preceding generations? In case this corpus of knowledge was still relevant, it could make the erosion lower.

Our interviewees indicated current and past uses. While there is no written trace of past uses of plants among the study groups, this study found that some elements (such as visits to their homeland and to their relatives who live there) are crucial for the maintenance of such LEK. We agree with the reviewer that the remembrance of some uses can be important, yet we also believe that such “passive” knowledge is not applied/applicable in food, medicinal and/or veterinary practices, and gets easily lost (see Kalle and Soukand, 2016).

14.- Replace ‘boarder’ with ‘border’.

Thank you, we have corrected it.

15.- The authors state that in some cases the uses are quoted by a small number of people. Related to this, and in general, I believe that calculating some indices (such as cultural value of informant’s consensus factor) would strengthen the analysis and the discussion of the results.

Thank you for pointing this out. This would be an interesting addition; however, we do not think that more indices would change the overall results we obtained from the application of the Jaccard index.

16.- In table 1, please specify the name of the local language in column 3’s heading.

We have corrected as requested.

17.- In table 1, three solutions appear for taxa undetermined at specific level, such as Allium ssp., Thymus sp., Nymphaea L. In the first case, I could understand several Allium species (but it is not clear, the local name being common). In the second case, I understand an undetermined Thymus species. In the third case, for Nymphaea, does this mean all species of the genus?

Thank you for noticing. Unfortunately, these three genera were identified via pictures, folk names, and plant descriptions, so we cannot further specify to the species level.

18.- In table 1, Artemisia ssp. should probably be replaced with Artemisia scoparia, and Tulip sp. should be replaced with Tulipa sp.

Thank you for noticing, we have corrected Tulipa sp. according to your suggestions, but we have kept Artemisia sp. because its identification was performed via pictures, folk names, and plant descriptions.   

Reviewer 2 Report

The manuscript entitled “Disadvantaged economic conditions and stricter border rules shape Afghan refugees’ ethnobotany. Insights from Kohat District, NW Pakistan”, authored by Shah et al. is an ethnobotanical study carried out in Kohat District (NW Pakistan). This study focuses on the use of food and medicinal plants (human and veterinary) by three groups, all of them belonging to the Pathans ethnic group.

The manuscript contains information provided by 90 informants (30 for each group) using free lists and semi-structured interviews. Global data are analyzed and comparisons are established between the different groups.

The manuscript is very well written and structured. The introduction clearly explains the historical and current context of the three groups. The objectives are clear and well defined.

The methodology is appropriate to achieve the proposed objectives. The results are clearly described, and followed by an interesting discussion, from which, despite containing some redundant information, some very sound conclusions are derived.

This manuscript represents a significant contribution to ethnobotany, but also to the study of migrant communities and the role of socioeconomic conditions in the erosion of traditional knowledge.

For all these reasons, we believe that the manuscript is suitable for publication in Plants, after minor revisions.

However, some minor issues should be addressed by the authors.

Line 43-44. Should the KW's be in alphabetical order? Replace "local knowledge" by "local ecological knowledge"

Line 63. “Ethnobotany” is not an appropriate term in this sentence.

Line 71. Aren't these different groups in Azerbaijan autochthonous? This sentence is a little confusing.

Line 116. The map legend is confusing. The range boundary is not clear and “Pathans “ (red flag) should be replaced by “Local Pathans” (in the whole manuscript).

Line 132. “willingly However” must be replaced by “willingly. However”

Lines 149-151. "The participants ranging from 50 to 85" does not agree with "middle-aged community".

Line 157. Table. “horticulturism” should be replaced by “horticulturalism”.

Lines 159-162. Afghans, Afghan Pathans and Pathans is confusing. It should be clarified to facilitate the read.

Lines 183-184. APG IV is more recent than the APG III.

Line 186. One space after "Excel." should be removed.

Line 194. “number” must be replaced by “Number”.

Line 202. Do the 63 families include the four mushrooms species? If not, the text should be "145 wild plant taxa belonging to 63 botanical families...".

Line 200-214. I don't understand how the percentages were calculated. Lines 206-208, the sum of the values is not 100. Table: how were they calculated? What is the total number of taxa in each case?

Line 247. Replace “Barberis” by “Berberis”.

Line 277. “energy production and galactagogue” are not really diseases. Galactagogue is an agent that promotes the secretion of milk.

Line 285. I don’t understand the meaning of “jiggery”.

Line 359. Apteranthes tuberculata not italicised.

Line 387. Are really 133 identified taxa? This result is not agree with the data shown in the Figure 4.

Line 395. This Figure should be improved. 

Lines 442-446. The paragraph showing the bias should be moved to the Material and Methods section.

Table 1 (supplementary material)

1.Achillea santolinoides Lag.,

7.Allium carolinianum Redouté,

14. Artemisia L.

37. Cydonia oblonga Mill.,

60. gastrointestinal pain, hepatitis C

77. Olea europaea subsp. cuspidata (Wall. & G.Don)

121. Tulipa sp.,

Author Response

Dear Reviewer,

Thank you for your meaningful feedback on our manuscript. We replied to each of the points you raised below. 

The manuscript entitled “Disadvantaged economic conditions and stricter border rules shape Afghan refugees’ ethnobotany. Insights from Kohat District, NW Pakistan”, authored by Shah et al. is an ethnobotanical study carried out in Kohat District (NW Pakistan). This study focuses on the use of food and medicinal plants (human and veterinary) by three groups, all of them belonging to the Pathans ethnic group.

The manuscript contains information provided by 90 informants (30 for each group) using free lists and semi-structured interviews. Global data are analyzed and comparisons are established between the different groups.

The manuscript is very well written and structured. The introduction clearly explains the historical and current context of the three groups. The objectives are clear and well defined.

The methodology is appropriate to achieve the proposed objectives. The results are clearly described, and followed by an interesting discussion, from which, despite containing some redundant information, some very sound conclusions are derived.

This manuscript represents a significant contribution to ethnobotany, but also to the study of migrant communities and the role of socioeconomic conditions in the erosion of traditional knowledge.

For all these reasons, we believe that the manuscript is suitable for publication in Plants, after minor revisions.

Thank you!

However, some minor issues should be addressed by the authors.

1- Line 43-44. Should the KW's be in alphabetical order? Replace "local knowledge" by "local ecological knowledge".

Thank you for noticing this, we have changed it.

2- Line 63. “Ethnobotany” is not an appropriate term in this sentence.

We have revised this sentence.

3- Line 71. Aren't these different groups in Azerbaijan autochthonous? This sentence is a little confusing.

We have clarified it.

4- Line 116. The map legend is confusing. The range boundary is not clear and “Pathans “ (red flag) should be replaced by “Local Pathans” (in the whole manuscript).

A new map has been uploaded.

5- Line 132. “willingly However” must be replaced by “willingly. However”

Thank you for noticing, we have changed it.

6- Lines 149-151. "The participants ranging from 50 to 85" does not agree with "middle-aged community".

We have revised the sentence accordingly.

7- Line 157. Table. “horticulturism” should be replaced by “horticulturalism”.

Thank you for noticing, we have corrected this.

8- Lines 159-162. Afghans, Afghan Pathans and Pathans is confusing. It should be clarified to facilitate the read.

Throughout the text the term “Pathans” was replaced by “local Pathans” while “Afghans” was replaced by “Afghan Pathans” to avoid confusion and facilitate reading.

9- Lines 183-184. APG IV is more recent than the APG III.

Thank you we have corrected this.

10- Line 186. One space after "Excel." should be removed.

Thank you for noticing, we have removed it.

11- Line 194. “number” must be replaced by “Number”.

Thank you for noticing, we have corrected this.

12- Line 202. Do the 63 families include the four mushrooms species? If not, the text should be "145 wild plant taxa belonging to 63 botanical families...".

Yes, we recorded the use of 145 wild plant taxa and four mushrooms belonging to 63 botanical and fungal families.

13- Line 200-214. I don't understand how the percentages were calculated. Lines 206-208, the sum of the values is not 100. Table: how were they calculated? What is the total number of taxa in each case?

In the current study the same species are utilized for multiple uses (food/vet/medicine) which is why the sum of the values is not 100. We have calculated the percentage as follows, i.e. total number of taxa reported for medicinal use / total number of taxa for any use.

14- Line 247. Replace “Barberis” by “Berberis”.

Thank you for noticing, we have corrected it.

15- Line 277. “energy production and galactagogue” are not really diseases. Galactagogue is an agent that promotes the secretion of milk.

Thank you for noticing, we have put the use into the “others” category.

16- Line 285. I don’t understand the meaning of “jiggery”.

We have corrected the term which is not “jiggery”, but “jaggery” (a traditional non-centrifugal cane sugar).

17- Line 359. Apteranthes tuberculata not italicised.

Thank you for noticing, we have corrected this.

18- Line 387. Are really 133 identified taxa? This result is not agree with the data shown in the Figure 4.

In figure 4 the Venn diagram comprised of all the identified taxa which equals 133, while we recorded 16 unidentified taxa but one unidentified species was mentioned by both AI and AO.

19- Line 395. This Figure should be improved. 

We thank the Reviewer for this comment, but we are unsure what specifically should be improved.

20- Lines 442-446. The paragraph showing the bias should be moved to the Material and Methods section.

Thank you, we have moved it to the Methods section.

21- Table 1 (supplementary material) 

  1. Achillea santolinoides Lag.,
  2. Allium carolinianum Redouté,
  3. Artemisia L.
  4. Cydonia oblonga Mill.,
  5. gastrointestinal pain, hepatitis C
  6. Olea europaea subsp. cuspidata (Wall. & G.Don)
  7. Tulipa sp.,

Thank you, we have corrected Table 1 (now Table 2) as suggested.
